# MGE-LDM: Joint Latent Diffusion for Simultaneous Music Generation and Source Extraction

**Yunkee Chae**[1]      **Kyogu Lee** [1,2,3]

Music and Audio Research Group (MARG)
[1] Interdisciplinary Program in Artificial Intelligence (IPAI)
[2] AIIS, [3] Department of Intelligence and Information
Seoul National University
{yunkimo95, kglee}@snu.ac.kr

## Abstract

We present **MGE-LDM**, a unified latent diffusion framework for simultaneous music generation, source imputation, and query-driven source separation. Unlike prior approaches constrained to fixed instrument classes, MGE-LDM learns a joint distribution over full mixtures, submixtures, and individual stems within a single compact latent diffusion model. At inference, MGE-LDM enables (1) complete mixture generation, (2) partial generation (i.e., source imputation), and (3) text-conditioned extraction of arbitrary sources. By formulating both separation and imputation as conditional inpainting tasks in the latent space, our approach supports flexible, class-agnostic manipulation of arbitrary instrument sources. Notably, MGE-LDM can be trained jointly across heterogeneous multi-track datasets (e.g., Slakh2100, MUSDB18, MoisesDB) without relying on predefined instrument categories. Audio samples are available at our project page [†].

## 1   Introduction

Recent advances in generative modeling have significantly accelerated progress in music audio synthesis, inspired by breakthroughs in the language and vision domains. Early autoregressive models such as WaveNet [1] demonstrated the feasibility of end-to-end waveform generation. Since then, two dominant approaches have emerged: (1) discrete-token models, which compress raw audio into quantized codes [2–5] for sequence modeling [6–8]; and (2) diffusion-based models [9–11], which synthesize audio by reversing a noise corruption process in the waveform domain [12, 13]. Building on this foundation, latent diffusion models (LDMs) [14], which operate in a compressed latent space, have delivered substantial gains in synthesis quality and have been successfully applied to music generation tasks [15–18].

Despite this progress, most music generation models produce a single, mixed waveform, lacking access to the individual instrument stems required for remixing, adaptive arrangement, or downstream production tasks. To recover these components, audio source separation techniques aim to decompose a mixture into its constituent tracks. Discriminative approaches [19–22] learn to directly regress each source from the input mixture, achieving strong performance. In contrast, generative separation models sample individual sources from a learned prior [23–27]. Recently, diffusion-based methods have demonstrated strong performance not only in speech separation [28, 29] but also in speech enhancement tasks [30–33], highlighting their potential for flexible, high-quality source recovery across audio domains.

---

[†]https://yoongi43.github.io/MGELDM_Samples/

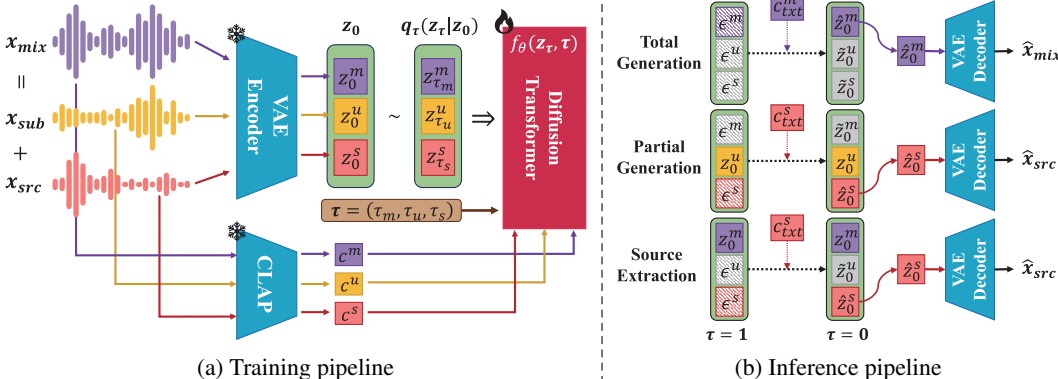

|  (a) Training pipeline | (b) Inference pipeline |

Figure 1: **Overview of MGE-LDM.** (a) *Training pipeline*: We train a three-track latent diffusion model on mixtures, submixtures, and sources. Each track is perturbed independently and conditioned on its corresponding timestep and CLAP embedding. The model is optimized using the v-objective, as detailed in Sections 3.2 and 3.4. (b) *Inference pipeline*: At test time, task-specific latents are either generated or inpainted based on available context and text prompts. The resulting latents are decoded into waveforms. See Section 3.3 for details.

Recent works have explored modeling the joint distribution of multi-track stems within a unified diffusion backbone, enabling mixture synthesis, accompaniment generation, and source separation within a single framework [34–36]. However, these approaches typically rely on predefined instrument classes for each track or assume that the mixture waveform is the linear sum of its constituent stems. While the additive assumption is valid in the waveform domain, it is incompatible with the nonlinear encoder–decoder structure of latent diffusion models, limiting the applicability of such methods in compressed latent spaces.

To address these limitations, we introduce **MGE-LDM**, a class-agnostic latent diffusion framework that jointly unifies music generation, partial generation, and arbitrary source extraction. Our approach models three interrelated latent variables: mixture, submixture, and source within a single diffusion backbone.

Given a waveform mixture $x^{(m)}$ and a chosen source $x^{(s)}$, we define the submixture as $x^{(u)} = x^{(m)} - x^{(s)}$. A pretrained encoder $E$ maps each waveform to its latent representation:

$$z^{(m)} = E(x^{(m)}), \quad z^{(u)} = E(x^{(u)}), \quad z^{(s)} = E(x^{(s)}). \tag{1}$$

We then train a diffusion model $f_\theta(z^{(m)}, z^{(u)}, z^{(s)})$ over this joint distribution.

At inference time, diffusion-based inpainting, conditioned on a subset of $\{z^{(m)}, z^{(u)}, z^{(s)}\}$, enables class-agnostic partial generation and text-driven source extraction without relying on fixed instrument vocabularies or linear mixing assumptions. A detailed formulation appears in Section 3.

Notably, our training paradigm is **dataset-agnostic**: by removing dependence on instrument labels, MGE-LDM can leverage all publicly available track-wise datasets in a single training run. For example, Slakh2100 [37] provides clean, isolated stems; MUSDB18 [38] and MoisesDB [39] include loosely labeled tracks such as `other` that aggregate multiple instruments. Existing frameworks [34–36, 40] typically ignore such labels, as they assume a one-to-one correspondence between sources and tracks, making it difficult to train and perform inference on mixtures containing sources outside predefined instrument classes.

In contrast, our three-track formulation (mixture, submixture, individual source) treats aggregated labels as submixtures. This design integrates them into joint training together with fully separated stems. This dataset flexibility simplifies data collection and enhances model robustness across heterogeneous source structures.

In summary, our contributions are as follows:

- We propose MGE-LDM, the first latent diffusion model to jointly address music generation, source imputation, and text-conditioned source extraction without relying on predefined instrument labels.

- We formulate source extraction as a conditional inpainting problem in the latent space, jointly modeling mixture, submixture, and source embeddings. This formulation enables class-agnostic, text-driven extraction of arbitrary stems within a unified latent diffusion framework.

- We propose a training strategy that assigns distinct diffusion timesteps to each of the mixture, submixture, and source tracks, allowing the model to learn adaptive score functions for diverse inpainting contexts.

- By eliminating dependence on fixed stem annotations, our framework ingests heterogeneous multi-track datasets (e.g., Slakh2100, MUSDB18, MoisesDB) in a unified manner, simplifying data aggregation and improving generalization to unseen instrumentation.

## 2   Related Work

**Audio Source Separation and Extraction.** Audio source separation aims to decompose a polyphonic mixture into its constituent tracks, while source extraction focuses on isolating a particular target sound, often guided by metadata, text prompts, or reference examples. Two dominant paradigms have emerged: discriminative models learn a direct mapping from the input mixture to each target stem via regression losses, either in the waveform domain or in spectrogram representations [19–22, 41, 42]. In contrast, generative approaches learn probabilistic priors over source distributions and recover individual stems via sampling [23–26, 43].

Recently, diffusion-based techniques have emerged as a powerful paradigm for audio decomposition, achieving strong results in both speech separation [28, 29] and enhancement [30–33]. These methods iteratively denoise a mixture under a learned score function, offering flexible and high-fidelity source recovery.

Query-based extraction further extends separation by conditioning the model on external cues such as class labels [44–46], visual signals [47, 48], or audio exemplars [49, 50]. Several studies have also demonstrated the effectiveness of natural language prompts for flexible, user-driven source isolation [48, 51–55]. In our framework, we employ the pretrained CLAP model [56] to obtain shared audio-text embeddings, enabling seamless, language-guided extraction of arbitrary stems within a multi-track latent diffusion architecture.

**Audio Generation Models.** Early neural audio synthesis methods focused on autoregressive architectures that model waveform dependencies sample by sample. WaveNet [1] demonstrated the effectiveness of dilated convolutions for end-to-end generation, while SampleRNN [57] extended this with hierarchical recurrence. Subsequent work adopted adversarial objectives to improve fidelity, using GANs to generate perceptually sharp outputs [58, 59].

Parallel efforts introduced discrete-token models, where audio is encoded into compact code sequences using vector quantization (e.g., VQ-VAE [2]). Jukebox [6] models long-range dependencies over codes using Transformers [60], while recent systems enhance fidelity through residual quantization [3–5] and hierarchical token modeling, where coarse-to-fine code representations are generated across multiple levels [7, 61–63]. MusicGen [8] improves decoding efficiency with delayed-token generation, and Instruct-MusicGen [64] extends it for targeted editing via instruction-tuned prompts. In parallel, token-based masked generative modeling techniques, originally developed for vision [65], have been adapted for audio, enabling efficient non-autoregressive synthesis and precise spectrogram inpainting [66, 67].

Diffusion-based generation emerged with DiffWave [12] and WaveGrad [13], which learn to iteratively denoise Gaussian-corrupted waveforms. These techniques have since been adapted for music-specific generation with structure and style conditioning [68, 69]. Latent diffusion models (LDMs) [14], which perform denoising in a compressed embedding space, have further advanced generation fidelity and scalability. LDM-based audio models such as AudioLDM [70, 71], MusicLDM [16], and Stable Audio [17, 18, 72] achieve state-of-the-art performance. Recent frameworks like AUDIT [73], InstructME [74] explore the use of diffusion for controllable and interactive audio editing.

**Multi-Track Music Audio Modeling.** Recent studies model multi-track music as a structured composition of interdependent stems. StemGen [75] employs an iterative, non-autoregressive transformer over discrete tokens to generate stems conditioned on text prompts. Jen-1 Composer [76] applies latent diffusion to jointly model four canonical stems (bass, drums, instrument, melody), producing

coherent multi-track compositions. MusicGen-Stem [77] combines per-stem vector quantization with an autoregressive decoder to synthesize bass, drums, and aggregated `other` components, and supports mixture-conditioned accompaniment generation. Diff-A-Riff [78] leverages latent diffusion to co-create stems complementary to given mixtures, later extended with diffusion transformers [79]. Other studies have similarly explored mixture-conditioned stem generation [80, 81].

A separate line of work focuses on joint modeling of synthesis and decomposition within a single diffusion backbone. Multi-Source Diffusion Models (MSDM) [34] model a fixed set of stems (`bass`, `drums`, `guitar`, and `piano`) within a shared diffusion framework, relying on an additive mixture assumption and a Dirac delta-based posterior sampler, following the EDM formulation for ODE-based sampling [82]. This line of work has since been extended in GMSDI [35], MSG-LD [36], and others [40, 83]. GMSDI enables variable-stem modeling and text-based conditioning but remains grounded in waveform-space additive mixing. MSG-LD adapts latent diffusion for four-stem modeling and classifier-free guidance [84], though it still assumes fixed instrument classes.

In contrast, our approach jointly models mixture, submixture, and source embeddings in latent space and casts both synthesis and arbitrary-source extraction as text-conditioned inpainting tasks, offering fully class-agnostic multi-track music processing without reliance on fixed instrument vocabularies or linear mixing assumptions.

## 3 Method

Figure 1 presents an overview of the proposed MGE-LDM framework. We train a three-track joint latent diffusion model that learns the distribution over the joint space of mixture, submixture, and source representations, as defined in Eq. (1). During inference, the model performs various tasks by exploiting the inpainting capability of diffusion models. We first describe how training triplets are constructed, then detail how they are used for joint diffusion-based training and inpainting-based inference.

### 3.1 Formulating Joint Latent Representation

Let $\{x_i\}_{i \in I}$ denote the set of time-domain audio stems, where the number of sources $|I|$ may vary across mixtures depending on their instrumentation. The mixture waveform is defined as:

$$x^{(m)} = \sum_{i \in I} x_i.$$

To construct a training example for our three-track model, we uniformly sample an index $j \in I$ and define:

$$x^{(s)} = x_j, \quad x^{(u)} = \sum_{i \in I \setminus \{j\}} x_i,$$

yielding the triplet $\left(x^{(m)}, x^{(u)}, x^{(s)}\right)$. We encode each element of this triplet using a pretrained variational autoencoder (VAE) [85] encoder $E$, resulting in latent representations $z^{(m)}, z^{(u)}, z^{(s)} \in \mathbb{R}^{C \times L}$, where $C$ and $L$ denote the latent channel and temporal dimensions, respectively. This formulation naturally accommodates mixtures with a variable number of stems. Regardless of the number of instruments present, any publicly available multi-track dataset can be decomposed into mixture, submixture, and source components for joint latent modeling.

### 3.2 Latent Diffusion Training with Three-Track Embeddings

We build upon the Stable Audio framework [72], employing a Diffusion Transformer (DiT) backbone [86] and training the model under the v-objective [87]. Below, we summarize the v-objective training procedure; full derivations and additional details are provided in Appendix A.1. Let the composite latent input be defined as:

$$\mathbf{z}_0 = \left(z_0^{(m)}, z_0^{(u)}, z_0^{(s)}\right) \in \mathbb{R}^{3 \times C \times L}$$

where $z_0^{(k)} \in \mathbb{R}^{C \times L}$ are (clean) track embeddings, with $k \in K = \{m, u, s\}$ denoting the track types – mixture, submixture, and source, respectively.

We aim to estimate the score $\nabla_{\mathbf{z}_\tau} \log q_\tau(\mathbf{z}_\tau)$ across continuous noise levels $\tau \in [\tau_{\min}, 1]$. To this end, we perturb the clean latent variable $\mathbf{z}_0$ with Gaussian noise, following:

$$\mathbf{z}_\tau = \alpha_\tau \mathbf{z}_0 + \beta_\tau \boldsymbol{\epsilon}, \quad \boldsymbol{\epsilon} \sim \mathcal{N}(\mathbf{0}, \mathbf{I}), \tag{2}$$

where the noise scaling coefficients are parameterized as:

$$\alpha_\tau = \cos(\phi_\tau), \quad \beta_\tau = \sin(\phi_\tau), \quad \phi_\tau = \frac{\pi}{2}\tau. \tag{3}$$

Here, $\tau \sim \mathcal{U}([\tau_{\min}, 1])$ is sampled from a truncated uniform distribution with $\tau_{\min} = 0.02$ for stability.

A denoising network $f_\theta(\mathbf{z}_\tau, \tau, \mathbf{c})$ is trained to estimate the score $\nabla_{\mathbf{z}_\tau} \log q_\tau(\mathbf{z}_\tau | \mathbf{c})$ using the v-objective:

$$\mathcal{L}(\theta) = \mathbb{E}_{\mathbf{z}_0, \boldsymbol{\epsilon}, \tau} \, ||f_\theta(\mathbf{z}_\tau, \tau, \mathbf{c}) - \boldsymbol{v}_\tau||_2^2, \quad \boldsymbol{v}_\tau = \frac{\partial \mathbf{z}_\tau}{\partial \phi_\tau} = \alpha_\tau \boldsymbol{\epsilon} - \beta_\tau \mathbf{z}_0. \tag{4}$$

The conditioning vector $\mathbf{c} = (c^{(m)}, c^{(u)}, c^{(s)})$ is derived using the audio branch of a pretrained CLAP encoder [56], applied to each component:

$$c^{(k)} = \text{CLAP}_{\text{audio}}(x^{(k)}), \quad \text{for } k \in K.$$

To enable classifier-free guidance (CFG) [84], each $c^{(k)}$ is independently dropped out with probability $p$ during training.

## 3.3 Inference via Conditional Sampling in Latent Space

In the image domain, *inpainting* refers to reconstructing missing or corrupted regions of an image by conditioning on surrounding pixels. Diffusion models have demonstrated strong zero-shot inpainting capabilities, enabling arbitrary mask completion without retraining [11, 88]. We extend this paradigm to the latent domain of music, operating over a joint distribution of mixture, submixture, and source embeddings. Downstream tasks are formulated as conditional generation problems, where known latents are treated as observed and unknown ones are sampled as missing components.

In all inference modes, we condition on natural-language queries using CLAP embeddings. When text conditioning is required, we use the text branch of CLAP to produce the prompt embedding $c^{(k)} = \text{CLAP}_{\text{text}}(c_{\text{text}}^{(k)})$, where $c_{\text{text}}^{(k)}$ is a free-form natural language description (e.g., "*the sound of an electric guitar*").

**Total Generation.** Let $p_\theta(z^{(m)}, z^{(u)}, z^{(s)})$ denote the implicit model distribution whose score we approximate with $f_\theta$. To synthesize a complete mixture, we condition only on the mixture prompt embedding $c^{(m)}$ or omit all conditions for unconditional generation. We sample the mixture latent $\hat{z}^{(m)}$ as:

$$\hat{z}^{(m)}, \tilde{z}^{(u)}, \tilde{z}^{(s)} \sim p_\theta(z^{(m)}, z^{(u)}, z^{(s)} | c^{(m)}, \varnothing^{(u)}, \varnothing^{(s)}), \tag{5}$$

where $\tilde{z}^{(u)}$ and $\tilde{z}^{(s)}$ are auxiliary latents that are discarded. Finally, the synthesized mixture waveform $\hat{x}^{(m)}$ is obtained by decoding $\hat{z}^{(m)}$ through the pretrained VAE decoder $D$:

$$\hat{x}^{(m)} = D(\hat{z}^{(m)}).$$

Hereafter, we use $\tilde{z}^{(k)}$ to denote any dummy latent that is not retained during inference.

**Partial Generation.** *Partial generation*, also known as *source imputation*, refers to the task of generating missing stems given partially observed sources. We approach this iteratively to progressively reconstruct the full mixture from the partial input.

Let $\bar{\mathcal{I}} = \{c_1, ..., c_J\}$ be an (ordered) set of CLAP-derived text embeddings, each corresponding to a target source description to be imputed. Let $x_0^{(u)}$ denote the waveform mixture of the observed sources, and let $z_0^{(u)} = E(x_0^{(u)})$ be its latent representation. We initialize the submixture latent with $z_0^{(u)}$ and generate each missing source sequentially.

At each step $j \in \{1, ..., J\}$, we sample a new source latent $\hat{z}_j^{(s)}$ conditioned on the current submixture and the text embedding $c_j^{(s)}$:

$$\tilde{z}_j^{(m)}, \hat{z}_j^{(s)} \sim p_\theta(z^{(m)}, z^{(s)} | z_{j-1}^{(u)}, \varnothing^{(m)}, \varnothing^{(u)}, c_j^{(s)}). \tag{6}$$

We then update the submixture by accumulating the decoded sources:

$$z_j^{(u)} = E\left(\sum_{l=0}^{j-1} D(\hat{z}_l^{(s)})\right).$$

After $J$ iterations, we obtain the full set of imputed sources $\{\hat{z}_j^{(s)}\}_{j=1}^{J}$ and reconstruct the final mixture waveform as:

$$x^{(m)} = x_0^{(u)} + \sum_{j=1}^{J} D(\hat{z}_j^{(s)}). \tag{7}$$

**Source Extraction.** Text-driven extraction of an arbitrary stem is performed by conditioning on a natural-language prompt. Given a prompt embedding $c^{(s)}$, we treat the mixture latent $z^{(m)}$ as observed and inpaint the submixture and source tracks:

$$\tilde{z}^{(u)}, \hat{z}^{(s)} \sim p_\theta(z^{(u)}, z^{(s)} \mid z^{(m)}, \varnothing^{(m)}, \varnothing^{(u)}, c^{(s)}), \tag{8}$$

where $\tilde{z}^{(u)}$ is an auxiliary prediction that is discarded. Finally, the isolated source waveform is reconstructed via $\hat{x}^{(s)} = D(\hat{z}^{(s)})$.

## 3.4 Track-aware Inpainting Model with Adaptive Timesteps

Conventional diffusion-based inpainting methods apply a uniform noise schedule across both observed and missing regions, failing to account for their differing uncertainty characteristics [9, 11, 88–90]. In the standard setup, a denoising model $f_\theta(\mathbf{z}_\tau, \tau)$ is trained to approximate the joint score of a perturbed latent variable.

Following the perturbation rule and v-objective from Section 3.2, the score estimate is expressed as:

$$\nabla_{\mathbf{z}_\tau} \log q_\tau(\mathbf{z}_\tau) \approx -\mathbf{z}_\tau - \frac{\alpha_\tau}{\beta_\tau} f_\theta(\mathbf{z}_\tau, \tau), \tag{9}$$

where $\alpha_\tau$ and $\beta_\tau$ is cosine schedule as defined in Eq. (3), following [87]. For notational simplicity, we omit the conditioning vectors $\mathbf{c}$ in this expression. A detailed derivation is provided in Appendix A.1.

Recently, region-aware adaptations of diffusion inpainting, including spatially varying noise schedules [91] and per-pixel timestep conditioning in TD-Paint [92], have demonstrated substantial improvements in semantic consistency by preserving fidelity in observed regions. Inspired by TD-Paint, we extend this idea to three-track music audio by assigning distinct timestep conditions to each track, thereby improving inpainting quality in the latent space.

We describe our track-wise adaptive timestep conditioned model using general notation. Let $K$ be a set of tracks, and let $N = |K|$ be the number of tracks. Define the clean latent tensor and the corresponding noise levels as:

$$\mathbf{z}_0 = (z_0^{(k)})_{k \in K} \in \mathbb{R}^{N \times C \times L}, \quad \boldsymbol{\tau} = (\tau_k)_{k \in K} \in [\tau_{\min}, 1]^N,$$

where each $\tau_k$ is either zero (for observed tracks) or equal to a shared sample $\tau \sim \mathcal{U}([\tau_{\min}, 1])$, depending on the inpainting configuration.

We define the track-wise product between a vector $x_{\boldsymbol{\tau}} \in \mathbb{R}^N$ and a latent tensor $\mathbf{z} \in \mathbb{R}^{N \times C \times L}$ as:

$$x_{\boldsymbol{\tau}} \odot \mathbf{z} := (x_{\boldsymbol{\tau}_k} z^{(k)})_{k \in K},$$

and extend this notation to the cosine noise schedule terms as follows:

$$\alpha_{\boldsymbol{\tau}} = (\alpha_{\tau_k})_{k \in K}, \quad \beta_{\boldsymbol{\tau}} = (\beta_{\tau_k})_{k \in K}.$$

We perturb the joint latent using track-wise noise:

$$\mathbf{z}_{\boldsymbol{\tau}} = \alpha_{\boldsymbol{\tau}} \odot \mathbf{z}_0 + \beta_{\boldsymbol{\tau}} \odot \boldsymbol{\epsilon}, \quad \boldsymbol{\epsilon} \sim \mathcal{N}(\mathbf{0}, \mathbf{I}), \tag{10}$$

where each track is independently scaled by its corresponding noise factor.

The denoiser $f_\theta(\mathbf{z_\tau}, \boldsymbol{\tau})$ is trained to regress the velocity target under the v-objective:

$$\boldsymbol{v}_{\boldsymbol{\tau}} = \alpha_{\boldsymbol{\tau}} \odot \boldsymbol{\epsilon} - \beta_{\boldsymbol{\tau}} \odot \mathbf{z}_0, \tag{11}$$

resulting in the following training loss:

$$\mathcal{L}(\theta) = \mathbb{E}_{\mathbf{z}_0, \boldsymbol{\epsilon}, \boldsymbol{\tau}} ||f_\theta(\mathbf{z_\tau}, \boldsymbol{\tau}) - \boldsymbol{v}_{\boldsymbol{\tau}}||_2^2. \tag{12}$$

In our setup, we use $N = 3$ with $K = \{m, u, s\}$, corresponding to the mixture, submixture, and source tracks, respectively. In practice, the loss is computed only over the unknown tracks.

During training, we first sample a noise level $\boldsymbol{\tau} \sim \mathcal{U}([\tau_{\min}, 1])$ and set the per-track timestep vector $\boldsymbol{\tau} \in \mathbb{R}^3$ according to one of the following four patterns:

$$\boldsymbol{\tau} \in \{(\tau, \tau, \tau), (0, \tau, \tau), (\tau, 0, \tau), (\tau, \tau, 0)\},$$

where each configuration is selected randomly for each training step.

Under this conditioning strategy, the full-noise setting $\boldsymbol{\tau} = (\tau, \tau, \tau)$ corresponds to learning the standard joint score, as described in Eq. (9). In contrast, a "single-zero" pattern allows the model to learn conditional score functions for the unobserved tracks while treating the others as fixed observations.

For example, when $(\tau_m, \tau_u, \tau_s) = (0, \tau, \tau)$, the model is trained to approximate the gradient:

$$\nabla_{(z_\tau^{(u)}, z_\tau^{(s)})} \log q_\tau(z_\tau^{(u)}, z_\tau^{(s)} | z_0^{(m)}), \tag{13}$$

which the joint-score formulation in Eq. (9) cannot compute in closed form. At inference time, we clamp the observed tracks by setting their noise levels to zero and apply standard reverse diffusion updates to the remaining (missing) tracks. Pseudocode and a detailed theoretical comparison with conventional inpainting methods are provided in Appendix B.

## 4 Experimental Setup

In this section, we outline our experimental protocol, including baseline models, datasets, and key implementation details. A comprehensive description of hyperparameters and training procedures is provided in Appendix D. All baseline results are re-evaluated using our test sets to ensure consistency with our experimental setup.

**Baselines.** We use two recent multi-track diffusion models — MSDM [34] and MSG-LD [36] — as baselines, both of which operate on a fixed set of stems: `bass`, `drums`, `guitar`, and `piano`. In addition to generative and inpainting performance, we assess source extraction capabilities against Hybrid Demucs (HDemucs) [20], which separates the mixture into `bass`, `drums`, `other`, and `vocals` stems, and AudioSep [53], which performs text-conditioned separation based on natural language queries.

**Datasets.** We train and evaluate on three multi-track music datasets: Slakh2100 [37], MUSDB18 [38] (denoted $\mathbf{M}_u$), and MoisesDB [39] (denoted $\mathbf{M}_o$). For Slakh2100, we define two subsets: $\mathbf{S}_A$, containing only `bass`, `drums`, `guitar`, and `piano` stems to match the MSDM and MSG-LD setup; and $\mathbf{S}_B$, which includes all remaining stems. Each dataset follows its predefined train/test split. We train our models on various dataset combinations to evaluate robustness under different source distributions and stem configurations.

**Implementation Details.** Our models use the Stable Audio backbone [72], which comprises an autoencoder and a DiT-based diffusion model. To better accommodate per-track variability in the joint latent space, we replace LayerNorm [93] with GroupNorm [94], using three groups to reflect the number of tracks.

To bridge the audio-text modality gap, we adopt stochastic linear interpolation between audio and text embeddings on the source track, following prior work on multimodal fusion [54, 95]. Concretely, we generate the prompt "*The sound of the* {`label`}" and compute the source conditioning vector $c^{(s)}$ as a convex combination of the CLAP text embedding and its corresponding audio embedding, where the interpolation weight $\alpha \sim \mathcal{U}([0, 1])$ is sampled randomly for each training example.

All of our models, except the one trained on the full dataset combination ($\mathbf{S}_A$, $\mathbf{S}_B$, $\mathbf{M}_u$, $\mathbf{M}_o$), are trained for 200K iterations with a batch size of 64, using 16 kHz audio segments of 10.24 seconds.

Table 1: **Total generation results.** Reported scores are FAD $\downarrow$, computed against mixture references from each test set. Values in parentheses indicate generation results conditioned on the prompt "*The sound of the bass, drums, guitar, and piano*", as detailed in Section 5.1. Among our models, $\mathcal{T}_1$ provides a fair comparison with baseline methods, as it is trained on the same dataset, while $\mathcal{T}_2$–$\mathcal{T}_4$ illustrate the effects of progressive dataset scaling. Bold values indicate the best results in each column, and underlined values denote the best results among models trained on the same $\mathbf{S}_A$ set.

| Model | | Train Set | | | | Test Set | | | |
|---|---|---|---|---|---|---|---|---|---|
| | | $\mathbf{S}_A$ | $\mathbf{S}_B$ | $\mathbf{M}_u$ | $\mathbf{M}_o$ | $\mathbf{S}_A$ | $\mathbf{S}_{\text{Full}}$ | $\mathbf{M}_u$ | $\mathbf{M}_o$ |
| MSDM | | ✓ | × | × | × | 4.21 | 6.04 | 7.92 | 7.41 |
| MSG-LD | | ✓ | × | × | × | 1.38 | 1.55 | 4.61 | 4.26 |
| MGE (ours) | $\mathcal{T}_1$ | ✓ | × | × | × | **0.47** (3.57) | 1.79 | 6.34 | 5.90 |
| | $\mathcal{T}_2$ | ✓ | ✓ | × | × | 3.14 (2.24) | **0.63** | 5.46 | 4.73 |
| | $\mathcal{T}_3$ | × | × | ✓ | ✓ | 8.80 (3.96) | 6.56 | 2.87 | 1.59 |
| | $\mathcal{T}_4$ | ✓ | ✓ | ✓ | ✓ | 6.83 (5.05) | 4.22 | **2.78** | **1.47** |

The full combination model is trained for 320K iterations with a batch size of 128. During sampling and inpainting, we apply classifier-free guidance (CFG) with a guidance scale of 2.0 and a per-track dropout probability of $p = 0.1$. All diffusion-based samples – including those from baseline models – are generated using 250 inference steps. We adopt DDIM sampling [96] for all our models, while each baseline uses its originally proposed sampling method.

## 5 Results

We evaluate MGE-LDM on three tasks: total generation, partial generation, and source extraction. Each result table indicates the training dataset(s) used and reports performance across multiple test sets. Unless otherwise specified, partial generation and source extraction are performed using text prompts of the form "*The sound of the* {label}." Abbreviations for all stem labels are listed in Appendix Table 4, and additional ablation results are provided in Appendix E.

### 5.1 Music Generation

Table 1 presents FAD (Fréchet Audio Distance) [97] scores computed using VGGish embeddings [98], a widely adopted metric for evaluating music generation quality. Note that the $\mathbf{S}_A$ test set contains only mixtures of `bass`, `drums`, `guitar`, and `piano`, while $\mathbf{S}_{\text{Full}}$ corresponds to the full Slakh2100 test set, which includes a broader and more diverse set of instruments.

We first compare our model $\mathcal{T}_1$ against MSDM and MSG-LD, where all models are trained on $\mathbf{S}_A$. Our model achieves the lowest FAD on the $\mathbf{S}_A$ test set, demonstrating superior fidelity in generating standard four-stem mixtures. However, its generalization to other test sets is more limited. On $\mathbf{S}_{\text{Full}}$, which includes a broader range of instruments, MSG-LD performs slightly better than $\mathcal{T}_1$, suggesting mild overfitting to the constrained training distribution. That said, $\mathcal{T}_1$ still outperforms MSDM across most test conditions.

To assess the effect of dataset extension, we next train on the full Slakh2100 dataset ($\mathbf{S}_A$+$\mathbf{S}_B$), resulting in model $\mathcal{T}_2$. This broader training improves performance on $\mathbf{S}_{\text{Full}}$ and enhances generalization compared to $\mathcal{T}_1$. However, $\mathcal{T}_2$ still lags behind MSG-LD on other subsets, and its performance on $\mathbf{S}_A$ degrades, likely due to increased variability in the training distribution.

The $\mathcal{T}_3$ model, trained on MUSDB18 ($\mathbf{M}_u$) and MoisesDB ($\mathbf{M}_o$), both containing real recordings rather than synthesized audio, achieves strong performance on their respective test sets. Despite the domain shift, it also performs competitively on Slakh2100, with results comparable to MSDM, highlighting the model's robustness to cross-domain generalization. Finally, our model trained on the combined dataset comprising Slakh2100, MUSDB18, and MoisesDB (denoted as $\mathcal{T}_4$), achieves the best overall results on both $\mathbf{M}_u$ and $\mathbf{M}_o$. It also outperforms $\mathcal{T}_3$ on $\mathbf{S}_{\text{Full}}$, benefiting from the broader training distribution.

We also report results in parentheses, which correspond to mixture generation conditioned on the text prompt "*The sound of the bass, drums, guitar, and piano.*" For models trained on datasets beyond $\mathbf{S}_A$,

Table 2: **Partial generation results.** Scores are reported using *sub*-FAD ↓, which measures the distance between the reference mixture and the sum of given and generated sources. Each column header (e.g., B, D, G) indicates the target source being generated, conditioned on the remaining stems. Bold and underlined values follow the same convention as in Table 1.

| Model | Train Set | | | | $S_A$ | | | | | | | | | | | | | | $S_B$ | | | | | | | |
|---|---|---|---|---|---|---|---|---|---|---|---|---|---|---|---|---|---|---|---|---|---|---|---|---|---|---|
| | $S_A$ | $S_B$ | $M_u$ | $M_o$ | B | D | G | P | BD | BG | BP | DG | DP | GP | BDG | BDP | BGP | DGP | Brs. | C.P. | Org. | Pipe | Reed | Str. | S.Lead | S.Pad |
| MSDM | ✓ | × | × | × | 0.56 | 1.06 | 0.49 | 0.70 | 2.23 | 1.56 | 1.95 | 1.64 | 1.83 | 2.31 | 3.09 | 3.53 | 5.72 | 3.86 | - | - | - | - | - | - | - | - |
| MSG-LD | ✓ | × | × | × | **0.33** | **0.34** | **0.49** | **0.48** | **0.70** | **1.08** | **1.05** | **0.86** | **0.83** | 1.47 | **1.43** | **1.42** | 2.31 | **1.76** | - | - | - | - | - | - | - | - |
| MGE (ours) $T_1$ | ✓ | × | × | × | 1.02 | 1.41 | 1.17 | 1.19 | 1.15 | 1.29 | 1.25 | 1.69 | 1.65 | **1.14** | 1.80 | 1.84 | **1.45** | 1.84 | **1.45** | 0.68 | 0.23 | 3.48 | 5.58 | 1.38 | 4.47 | 1.08 |
| $T_2$ | ✓ | ✓ | × | × | 2.11 | 2.99 | 1.99 | 2.74 | 4.07 | 2.32 | 4.18 | 3.54 | 3.90 | 3.18 | 4.93 | 5.69 | 4.25 | 4.66 | 5.96 | **0.41** | 1.03 | 3.66 | 3.52 | 2.79 | 0.88 | 2.32 |
| $T_3$ | × | × | ✓ | ✓ | 1.43 | 1.29 | 3.34 | 2.30 | 1.85 | 3.64 | 2.83 | 2.95 | 2.36 | 4.39 | 3.30 | 3.57 | 6.03 | 3.86 | 3.58 | 0.58 | **0.15** | 0.22 | **0.56** | **0.61** | 0.54 | 0.48 |
| $T_4$ | ✓ | ✓ | ✓ | ✓ | 1.14 | 1.50 | 3.75 | 2.47 | 2.06 | 4.06 | 2.82 | 3.37 | 2.74 | 4.55 | 3.94 | 4.05 | 5.66 | 4.06 | 5.09 | 0.42 | 0.56 | **0.20** | 3.14 | 3.95 | **0.31** | **0.40** |

Table 3: **Source extraction results.** Metrics are reported as Log-Mel L1 distance ↓. For baseline models, scores are shown only for stems included in their fixed output set. Bold and underlined values follow the same convention as in Table 1.

| Model | Train Set | | | | $S_A$ | | | | $S_B$ | | | | | | | | $M_u$ | | | $M_o$ | | | | | | |
|---|---|---|---|---|---|---|---|---|---|---|---|---|---|---|---|---|---|---|---|---|---|---|---|---|---|---|
| | $S_A$ | $S_B$ | $M_u$ | $M_o$ | B | D | G | P | Brs. | C.P. | Org. | Pipe | Reed | Str. | S.Lead | S.Pad | V | B | D | V | B | D | G | P | B.Str | Perc. |
| HDemucs | × | × | ✓ | × | 1.49 | 0.90 | - | - | - | - | - | - | - | - | - | - | **1.50** | 1.99 | 1.53 | **0.83** | 1.71 | 1.10 | - | - | - | - |
| AudioSep | × | × | × | × | 2.36 | 1.67 | 3.41 | 2.42 | **3.13** | 2.84 | 3.26 | 3.04 | 3.15 | 2.57 | 2.8 | 2.06 | 2.66 | 4.07 | 1.89 | 1.54 | 3.37 | 1.31 | 1.42 | 1.70 | **2.36** | |
| MSDM | ✓ | × | × | × | 1.90 | 1.51 | 3.32 | 2.70 | - | - | - | - | - | - | - | - | - | 2.56 | 1.69 | - | 2.15 | 1.31 | **1.28** | **1.51** | - | - |
| MSG-LD | ✓ | × | × | × | **1.20** | 1.24 | 2.24 | 1.85 | - | - | - | - | - | - | - | - | - | 1.96 | 1.60 | - | 1.72 | 1.49 | 2.36 | 2.06 | - | - |
| MGE (ours) $T_1$ | ✓ | × | × | × | 1.28 | **0.66** | **1.27** | **1.07** | 3.22 | 3.07 | 3.13 | 3.11 | 3.30 | 2.77 | 2.68 | 2.30 | 3.80 | **1.91** | **1.33** | 5.15 | **1.61** | **1.10** | 2.86 | 2.68 | 2.03 | 2.94 |
| $T_2$ | ✓ | ✓ | × | × | 1.68 | 2.71 | 2.69 | 2.16 | 3.43 | **2.16** | **1.84** | 2.33 | 3.07 | 2.44 | **2.31** | **1.93** | 3.55 | 2.14 | 2.15 | 4.66 | 1.86 | 2.11 | 2.28 | 2.18 | 1.93 | **2.36** |
| $T_3$ | × | × | ✓ | ✓ | 1.80 | 0.99 | 2.89 | 2.01 | 3.17 | 2.51 | 3.61 | 2.13 | 2.86 | 2.22 | 2.78 | 2.22 | 1.85 | **1.56** | 1.17 | 0.98 | 1.10 | 0.90 | 1.04 | 1.58 | **1.62** | 2.49 |
| $T_4$ | ✓ | ✓ | ✓ | ✓ | 1.67 | 0.83 | 2.61 | 1.77 | 3.15 | 2.29 | 2.22 | **1.95** | **2.61** | **1.85** | 2.71 | 3.68 | 1.76 | **1.56** | **1.13** | 1.01 | **1.07** | **0.86** | **1.02** | **1.40** | 2.25 | 2.69 |

including $T_2$, $T_3$, and $T_4$, the generated mixtures generally contain a wider variety of instruments. This can result in a distributional mismatch with the $S_A$ test set, which contains only `bass`, `drums`, `guitar`, and `piano`, thereby increasing the FAD due to reference-target discrepancy. To mitigate this, we apply text conditioning at inference time using the above prompt to constrain the generated mixture to match the reference instrument set. As shown in the results, this conditioning significantly improves FAD for $T_2$, $T_3$, and $T_4$, yielding performance comparable to the MSDM baseline.

Table 2 presents results for partial generation, evaluated using the *sub*-FAD metric. This metric computes the FAD between the original mixture and a reconstruction formed by combining the given submixture with the generated stems. This evaluation protocol was introduced in prior work on music stem completion [34, 36, 63].

On $S_A$, our model $T_1$ performs worse than MSG-LD for single-source imputation but shows competitive or better performance as the number of generated stems increases. Among our models, $T_1$ performs best on $S_A$ due to domain alignment. However, for imputation tasks involving broader instrument classes in $S_B$, models trained on more diverse datasets perform better. Notably, the $T_3$ model, trained solely on $M_u$ and $M_o$, achieves strong results on $S_B$ despite not being exposed to Slakh2100 during training. In particular, it performs well on instruments such as `organ`, `pipe`, and `strings`. The fully trained model $T_4$ further improves performance across several $S_B$ stems, including `synth lead`, `synth pad`, and `pipe`.

## 5.2 Source Extraction

We evaluate text-queried source extraction using the Log Mel L1 distance, following MSG-LD [36], due to the inherent phase mismatch in latent-domain models that complicates waveform-domain evaluation. Table 3 presents results for a variety of stems across the $S_A$, $S_B$, $M_u$, and $M_o$ test sets. Alongside MSDM and MSG-LD, we compare with two additional baselines: HDemucs [20], a strong waveform- and spectrogram-domain separation model trained on fixed stems; and AudioSep [53], a recent system that enables natural language-driven extraction. For fixed-stem baselines (e.g., MSDM, MSG-LD, HDemucs), we report both in-distribution results and out-of-distribution generalization where possible (e.g., `bass` in $M_u$).

Our model $T_1$, trained solely on $S_A$, performs strongly on the canonical Slakh stems (`bass`, `drums`, `guitar`, `piano`), outperforming MSG-LD on all but `bass`. However, it generalizes poorly to less common stems and real-world recordings. By expanding the training set to encompass the full

Slakh2100 dataset, model $\mathcal{T}_2$ achieves improved performance in categories such as `chromatic percussion`, `organ`, `synth lead`, and `synth pad`, demonstrating the importance of broader intra-domain coverage.

Interestingly, model $\mathcal{T}_3$ generalizes competitively with $\mathcal{T}_2$ to synthetic stems, even outperforming some stems such as `drums`. This indicates cross-domain robustness in our latent inpainting formulation. Finally, model $\mathcal{T}_4$, trained on the combined dataset, exhibits robust performance across both synthetic and real-world domains. It maintains strong results on Slakh2100, while achieving the lowest Log Mel L1 scores across most stems in $\mathbf{M}_u$ and $\mathbf{M}_o$. This highlights that incorporating synthetic data such as Slakh2100 alongside real recordings can enhance generalization and improve separation quality on real-world audio.

Overall, MGE-LDM delivers robust, class-agnostic extraction with strong performance across a wide range of instrument types and recording domains, highlighting its effectiveness for text-driven music source extraction in both synthetic and real-world settings.

## 6 Limitations

While MGE-LDM provides a flexible, class-agnostic framework for multi-track audio modeling, several limitations remain. First, all experiments are conducted using 16 kHz monaural audio, which constrains upper-frequency resolution and omits spatial cues, thereby limiting realism for high-fidelity or stereo music applications. Second, the model relies on CLAP-based semantic conditioning, which introduces a modality gap between text and audio [99]. This can occasionally lead to semantic drift during extraction, resulting in irrelevant or hallucinated sources, particularly for stems with limited training data.

Third, although MGE-LDM reduces dependence on fixed instrument classes, it still requires multi-stem supervision during training. This dependency restricts applicability to fully unlabeled or large-scale web audio collections. Fourth, training on MUSDB18 alone with the same number of iterations as other configurations leads to overfitting, likely due to the limited duration (approximately 10 hours) of its training split. This highlights the challenge of achieving robust performance in low-resource multi-track settings.

Finally, our model is trained using triplets $(\textit{mix}, \textit{submix}, \textit{source})$ that satisfy $\textit{mix} = \textit{submix} + \textit{source}$ in waveform space; however, the latent diffusion process does not enforce an explicit additivity constraint for generated triplets. We believe this omission contributes directly to hallucination phenomena, where the model extracts sources absent from the mixture. Postolache et al. [27] addresses a related issue by enforcing additivity in a discrete VQ-VAE latent space, estimating the joint likelihood of two sources by counting codebook co-occurrences, effectively modeling $p(z_{\text{mix}}|z_{\text{src}_1}, z_{\text{src}_2})$, where $z_*$ are quantized latent codes. Our current pipeline, however, operates in a continuous latent space, which precludes the direct use of such discrete bin-counting methods. Adapting this latent-domain likelihood formulation to continuous spaces, for example, by designing suitable regularizers or adopting a VQ-VAE-based encoder with discrete diffusion [100–102] or MaskGiT [65]-style generation, represents a promising direction for future work.

## 7 Conclusion

We have presented MGE-LDM, a unified class-agnostic latent diffusion framework that jointly models mixtures, submixtures, and individual sources for music generation, stem completion, and text-driven extraction. By formulating stem completion and source extraction as conditional inpainting in a shared latent space and by introducing track-dependent timestep conditioning, we overcome the limitations of fixed-class, additive mixing assumptions and achieve flexible manipulation of arbitrary instrument tracks. Empirically, MGE-LDM matches or exceeds specialized baselines on Slakh2100 generation and separation benchmarks, while uniquely supporting zero-shot, language-guided extraction across heterogeneous multi-track datasets.

## Acknowledgements

This work was supported by the National Research Foundation of Korea (NRF) grant funded by the Korea government (MSIT) [No. RS-2024-00461617, 90%], Information & communications Technology Planning & Evaluation (IITP) grant funded by the Korea government (MSIT) [No. RS-2021-II211343, Artificial Intelligence Graduate School Program (Seoul National University), 10%] Additionally, the GPUs were partly supported by the National IT Industry Promotion Agency (NIPA)'s high-performance computing support program in 2025.

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

# A  Background

In this section, we review three foundational components that support our method: (1) the v-objective score-matching formulation for diffusion models [87], which underpins our latent modeling, and (2) the canonical inpainting algorithm introduced by Song et al. [11] and (3) further refined by Lugmayr et al. [89].

## A.1  v-Objective Diffusion

We adopt the continuous-time diffusion framework proposed by Salimans et al. [87], which has been widely applied in recent music generation models [15, 17, 18, 72].

The forward perturbation kernel is defined as:

$$q(\mathbf{z}_\tau|\mathbf{z}_0) = \mathcal{N}(\mathbf{z}_\tau; \alpha_\tau \mathbf{z}_0, \beta_\tau^2 \mathbf{I}) \tag{14}$$

where:

$$\alpha_\tau = \cos(\phi_\tau), \quad \beta_\tau = \sin(\phi_\tau), \quad \phi_\tau = \frac{\pi}{2}\tau, \quad \tau \in [0, 1].$$

Instead of predicting the noise $\epsilon$ as in DDPMs [10], the v-objective formulation introduces a *velocity* target:

$$\boldsymbol{v}_\tau = \frac{\partial \mathbf{z}_\tau}{\partial \phi_\tau} = \frac{\partial \cos(\phi_\tau)}{\partial \phi_\tau}\mathbf{z}_0 + \frac{\partial \sin(\phi_\tau)}{\partial \phi_\tau}\epsilon = -\sin(\phi_\tau)\mathbf{z}_0 + \cos(\phi_\tau)\epsilon = \alpha_\tau\epsilon - \beta_\tau\mathbf{z}_0, \tag{15}$$

and trains the denoiser $f_\theta(\mathbf{z}_\tau, \tau)$ to directly regress this velocity.

**Recovering $\mathbf{z}_0$ and $\epsilon$.** Using the definition of $\boldsymbol{v}_\tau$ and recalling the perturbation process defined in Eq. (2),

$$\mathbf{z}_\tau = \alpha_\tau\mathbf{z}_0 + \beta_\tau\epsilon \tag{16}$$

we can rearrange the terms to recover the clean latent by:

$$\sin(\phi_\tau)\mathbf{z}_0 = \cos(\phi_\tau)\epsilon - \boldsymbol{v}_\tau$$
$$= \frac{\cos(\phi_\tau)}{\sin(\phi_\tau)}(\mathbf{z}_\tau - \cos(\phi_\tau)\mathbf{z}_0) - \boldsymbol{v}_\tau,$$
$$\sin^2(\phi_\tau)\mathbf{z}_0 = \cos(\phi_\tau)\mathbf{z}_\tau - \cos^2(\phi_\tau)\mathbf{z}_0 - \sin(\phi_\tau)\boldsymbol{v}_\tau,$$
$$(\sin^2(\phi_\tau) + \cos^2(\phi_\tau))\mathbf{z}_0 = \mathbf{z}_0 = \cos(\phi_\tau)\mathbf{z}_\tau - \sin(\phi_\tau)\boldsymbol{v}_\tau,$$

thus we get

$$\mathbf{z}_0 = \alpha_\tau\mathbf{z}_\tau - \beta_\tau\boldsymbol{v}_\tau. \tag{17}$$

Similarly, the noise vector can be expressed as:

$$\epsilon = \beta_\tau\mathbf{z}_\tau + \alpha_\tau\boldsymbol{v}_\tau \tag{18}$$

**Score function approximation.** The marginal score $\nabla_{\mathbf{z}_\tau} \log q_{\mathbf{z}_\tau}(\mathbf{z}_\tau)$ at timestep (or noise level) $\tau$ is approximated as:

$$\nabla_{\mathbf{z}_\tau} \log q_\tau(\mathbf{z}_\tau) \approx \frac{\alpha_\tau\hat{\mathbf{z}}_\theta(\mathbf{z}_\tau) - \mathbf{z}_\tau}{\beta_\tau^2}$$
$$= \frac{\alpha_\tau^2\mathbf{z}_\tau - \alpha_\tau\beta_\tau f_\theta(\mathbf{z}_\tau, \tau) - \mathbf{z}_\tau}{\beta_\tau^2}$$
$$= \frac{-\beta_\tau^2\mathbf{z}_\tau - \alpha_\tau\beta_\tau f_\theta(\mathbf{z}_\tau, \tau)}{\beta_\tau^2}$$
$$= -\mathbf{z}_\tau - \frac{\alpha_\tau}{\beta_\tau}f_\theta(\mathbf{z}_\tau, \tau),$$

where $\hat{\mathbf{z}}_\theta(\mathbf{z}_\tau) = \alpha_\tau\mathbf{z}_\tau - \beta_\tau f_\theta(\mathbf{z}_\tau, \tau)$, from the Eq. (17).

**DDIM sampling.** We adopt the DDIM sampling [96] to generate samples in a non-stochastic, deterministic manner. Given a latent $\mathbf{z}_\tau$, each reverse step is computed as:

$$\hat{\boldsymbol{v}}_\tau = f_\theta(\mathbf{z}_\tau, \tau)$$
$$\hat{\mathbf{z}}_0 = \alpha_\tau\mathbf{z}_\tau - \beta_\tau\hat{\boldsymbol{v}}_\tau$$
$$\hat{\epsilon}_\tau = \beta_\tau\mathbf{z}_\tau + \alpha_\tau\hat{\boldsymbol{v}}_\tau$$
$$\mathbf{z}_{\tau'} = \alpha_{\tau'}\hat{\mathbf{z}}_0 + \beta_{\tau'}\hat{\epsilon}_\tau,$$

where $\tau' < \tau$ is taken from a linearly spaced decreasing schedule from 1 to 0. We use 250 inference steps in all experiments.

## A.2 Canonical Inpainting in Score-Based Models

The core idea behind inpainting in score-based generative models is to estimate the score of the unknown region conditioned on the known region [11, 34].

Let K denote the set of all tracks. Suppose a subset $\Omega \subset K$ is observed (i.e., known), and let $\Gamma = K \setminus \Omega$ denote the complement, i.e., the unobserved tracks we aim to inpaint. Define $\mathbf{z}^\Omega := \{z^{(k)}\}_{k \in \Omega}$ and $\mathbf{z}^\Gamma := \{z^{(k)}\}_{k \in \Gamma}$. The goal is to approximate the conditional score:

$$\nabla_{\mathbf{z}_\tau^\Gamma} \log q_\tau(\mathbf{z}_\tau^\Gamma | \mathbf{z}_0^\Omega). \tag{19}$$

This conditional gradient is generally intractable for a score model trained only on joint marginals. However, following Song et al. [11], we can approximate it via:

$$
\begin{aligned}
q_\tau(\mathbf{z}_\tau^\Gamma | \mathbf{z}_0^\Omega) &= \int q_\tau(\mathbf{z}_\tau^\Gamma, \mathbf{z}_\tau^\Omega | \mathbf{z}_0^\Omega) d\mathbf{z}_\tau^\Omega \\
&= \int q_\tau(\mathbf{z}_\tau^\Gamma | \mathbf{z}_\tau^\Omega, \mathbf{z}_0^\Omega) q_\tau(\mathbf{z}_\tau^\Omega | \mathbf{z}_0^\Omega) d\mathbf{z}_\tau^\Omega \\
&= \mathbb{E}_{q_\tau(\mathbf{z}_\tau^\Omega | \mathbf{z}_0^\Omega)} \left[ q_\tau(\mathbf{z}_\tau^\Gamma | \mathbf{z}_\tau^\Omega, \mathbf{z}_0^\Omega) \right] \\
&\approx \mathbb{E}_{q_\tau(\mathbf{z}_\tau^\Omega | \mathbf{z}_0^\Omega)} \left[ q_\tau(\mathbf{z}_\tau^\Gamma | \mathbf{z}_\tau^\Omega) \right] \tag{20} \\
&\approx q_\tau(\mathbf{z}_\tau^\Gamma | \hat{\mathbf{z}}_\tau^\Omega), \tag{21}
\end{aligned}
$$

where $\hat{\mathbf{z}}_\tau^\Omega \sim q_\tau(\mathbf{z}_\tau^\Omega | \mathbf{z}_0^\Omega) = \mathcal{N}(\mathbf{z}_\tau^\Omega; \alpha_\tau \mathbf{z}_0^\Omega, \beta_\tau^2 \mathbf{I})$ is a noised sample of the known region. Accordingly, the conditional score can be approximated as:

$$
\begin{aligned}
\nabla_{\mathbf{z}_\tau^\Gamma} \log q_\tau(\mathbf{z}_\tau^\Gamma | \mathbf{z}_0^\Omega) &\approx \nabla_{\mathbf{z}_\tau^\Gamma} \log q_\tau(\mathbf{z}_\tau^\Gamma | \hat{\mathbf{z}}_\tau^\Omega) \\
&= \nabla_{\mathbf{z}_\tau^\Gamma} \log q_\tau([\mathbf{z}_\tau^\Gamma; \hat{\mathbf{z}}_\tau^\Omega]),
\end{aligned}
$$

where $[\mathbf{z}_\tau^\Gamma; \hat{\mathbf{z}}_\tau^\Omega]$ denotes a composite latent vector, such that the known region is replaced by $\hat{\mathbf{z}}_\tau^\Omega$, while the unknown region remains as $\mathbf{z}_\tau^\Gamma$, adopting the same notation as Song et al. [11].

This approximation enables zero-shot inpainting without requiring retraining: at each diffusion timestep, a noised version of the known latents is sampled, concatenated with the current estimate of the unknown latents, and passed to the score model. The resulting gradient is then applied to update only the unknown region. This process is repeated throughout the reverse diffusion trajectory.

## A.3 RePaint

Lugmayr et al. proposed *RePaint* [89], a resampling-based mechanism that improves score-based inpainting by repeating the diffusion process across multiple forward–reverse cycles. Their key insight is that, in conventional inpainting (as described in Eq. (21)), the sampled noise for the known region is independent of the generated (inpainted) region. This lack of synchronization can lead to semantic inconsistencies and disharmony between the known and unknown parts of the sample.

To address this, RePaint introduces a resampling mechanism during generation. At each denoising timestep, the algorithm alternates between one reverse diffusion step and one forward diffusion step, repeating this cycle $U$ times. These micro-steps refine the sampling distribution and can have the effect of partially marginalizing over the known region at noise level $\tau$ in Equation (21), thereby reducing the approximation error inherent in conditional score estimation. This iterative resampling procedure improves consistency but incurs a higher computational cost, as each denoising step requires multiple forward–reverse passes, making RePaint significantly more expensive than standard inpainting methods.

While originally proposed for DDPM-based models, RePaint can be adapted to velocity-based objectives as used in our framework. We apply this adaptation in the sampling procedure described in Algorithm 1. Setting the resampling count $U = 1$ recovers the canonical single-sample inpainting method described in Appendix A.2.

# B Adaptive Timestep Conditioning and Score Approximation

Conventional inpainting approaches approximate the conditional score in Eq. (19) using a single sampled estimate of the known latents. This corresponds to a high-variance Monte Carlo estimate of the expectation over $q_\tau(\mathbf{z}_\tau^\Omega | \mathbf{z}_0^\Omega)$, which may lead to instability — especially at high noise levels.

By contrast, the adaptive timestep model proposed in Section 3.4, inspired by TD-Paint [92], circumvents this marginalization by training the model to directly approximate the conditional score.

---

**Algorithm 1** Inpainting using the RePaint approach.

---

**Input:**
Number of timesteps $T$;
Re-denoising steps per reverse step $U$;
Noise schedule $\{\tau_i\}_{i=0}^T$ with $\alpha_\tau, \beta_\tau$;
Binary mask $m$ (1 for known, 0 for unknown);
Known clean (masked) latents $\mathbf{z}_0^{\text{known}}$;
Denoiser network $f_\theta$

1: $\mathbf{z}_{\tau_T} \sim \mathcal{N}(\mathbf{0}, I)$
2: **for** $i = T, \dots, 1$ **do**
3:     **for** $u = 1, \dots, U$ **do**
4:                                          ▷ DDIM sampling step
5:         $\hat{\boldsymbol{v}}_{\tau_i} \leftarrow f_\theta(\mathbf{z}_{\tau_i}, \tau_i)$
6:         $\hat{\mathbf{z}}_0 \leftarrow \alpha_{\tau_i}\mathbf{z}_{\tau_i} - \beta_{\tau_i}\hat{\boldsymbol{v}}_{\tau_i}$
7:         $\hat{\boldsymbol{\epsilon}} \leftarrow \beta_{\tau_i}\mathbf{z}_{\tau_i} + \alpha_{\tau_i}\hat{\boldsymbol{v}}_{\tau_i}$
8:         $\hat{\mathbf{z}}_{\tau_{i-1}}^{\text{unknown}} \leftarrow \alpha_{\tau_{i-1}}\hat{\mathbf{z}}_0 + \beta_{\tau_{i-1}}\hat{\boldsymbol{\epsilon}}$
9:                                      ▷ Sample the known regions
10:        $\boldsymbol{\epsilon} \sim \mathcal{N}(\mathbf{0}, I)$ if $i > 1$, else 0
11:        $\mathbf{z}_{\tau_{i-1}}^{\text{known}} \leftarrow \alpha_{\tau_{i-1}}\mathbf{z}_0^{\text{known}} + \beta_{\tau_{i-1}}\boldsymbol{\epsilon}$
12:                              ▷ Combine known and generated regions
13:        $\mathbf{z}_{\tau_{i-1}} \leftarrow m \odot \mathbf{z}_{\tau_{i-1}}^{\text{known}} + (1 - m) \odot \hat{\mathbf{z}}_{\tau_{i-1}}^{\text{unknown}}$
14:                                        ▷ Reapply forward process
15:        **if** $u < U$ and $t > 1$ **then**
16:            $\mathbf{z}_{\tau_i} \sim \mathcal{N}\left(\frac{\alpha_{\tau_i}}{\alpha_{\tau_{i-1}}}\mathbf{z}_{\tau_{i-1}}, \left(1 - \frac{\alpha_{\tau_i}^2}{\alpha_{\tau_{i-1}}^2}\right) I\right)$
17:        **end if**
18:     **end for**
19: **end for**
20: **return** $\mathbf{z}_{\tau_0}$

---

Following the notation in Section 3.4 and Appendix A.2, we assume $\mathbf{z}_0 = [\mathbf{z}_0^\Gamma; \mathbf{z}_0^\Omega]$ is a clean sample from the dataset. Then, using an alternative factorization:

$$q_\tau(\mathbf{z}_\tau^\Gamma|\mathbf{z}_0^\Omega) = \int q_\tau(\mathbf{z}_\tau^\Gamma, \mathbf{x}_0^\Gamma|\mathbf{z}_0^\Omega)d\mathbf{x}_0^\Gamma$$

$$= \int q_\tau(\mathbf{z}_\tau^\Gamma|\mathbf{x}_0^\Gamma, \mathbf{z}_0^\Omega)q(\mathbf{x}_0^\Gamma|\mathbf{z}_0^\Omega)d\mathbf{x}_0^\Gamma$$

$$= \mathbb{E}_{q(\mathbf{x}_0^\Gamma|\mathbf{z}_0^\Omega)}\left[q_\tau(\mathbf{z}_\tau^\Gamma|\mathbf{x}_0^\Gamma, \mathbf{z}_0^\Omega)\right] \tag{22}$$

$$\approx q_\tau(\mathbf{z}_\tau^\Gamma|\mathbf{z}_0^\Gamma, \mathbf{z}_0^\Omega) = q_\tau(\mathbf{z}_\tau^\Gamma|\mathbf{z}_0^\Gamma) \tag{23}$$

$$= \mathcal{N}(\mathbf{z}_\tau^\Gamma; \alpha_\tau\mathbf{z}_0^\Gamma, \beta_\tau^2\mathbf{I}), \tag{24}$$

where the approximation assumes that $\mathbf{z}_0^\Gamma \sim q(\mathbf{x}_0^\Gamma|\mathbf{z}_0^\Omega)$ is available from the dataset. Unlike the marginalization-based approximation in Eq. (21), this expression introduces no sampling noise during inference, thereby reducing variance.

From this, the conditional score can be written as:

$$\nabla_{\mathbf{z}_\tau^\Gamma} \log q_\tau(\mathbf{z}_\tau^\Gamma|\mathbf{z}_0^\Omega) \approx \frac{\alpha_\tau\mathbf{z}_0^\Gamma - \mathbf{z}_\tau^\Gamma}{\beta_\tau^2} \tag{25}$$

$$\approx \frac{\alpha_\tau\hat{\mathbf{z}}_\theta(\mathbf{z}_\tau^\Gamma, \mathbf{z}_0^\Omega, \boldsymbol{\tau}^\Gamma) - \mathbf{z}_\tau^\Gamma}{\beta_\tau^2}, \tag{26}$$

$$= -\mathbf{z}_\tau^\Gamma - \frac{\alpha_\tau}{\beta_\tau}f_\theta(\mathbf{z}_\tau^\Gamma, \mathbf{z}_0^\Omega, \boldsymbol{\tau}^\Gamma)_\Gamma, \tag{27}$$

where $f_\theta(\cdot)_\Gamma$ denotes the output corresponding to the unknown region. The per-track timestep vector $\boldsymbol{\tau}^\Gamma$ is defined as:

$$\boldsymbol{\tau}_k^\Gamma = \begin{cases} \tau, & \text{if } k \in \Gamma \\ 0, & \text{if } k \in \Omega \end{cases} \quad \text{for each } k \in K, \tag{28}$$

---

**Algorithm 2** Inpainting using adaptive timestep approach

---

**Input:**
    Number of timesteps $T$;
    Re-denoising steps per reverse step $U$;
    Noise schedule $\{\tau_i\}_{i=0}^T$ with $\alpha_\tau, \beta_\tau$;
    Binary mask $m$ (1 for known, 0 for unknown);
    Known clean (masked) latents $\mathbf{z}_0^{\text{known}}$;
    Denoiser network $f_\theta$

1: $\boldsymbol{\epsilon} \sim \mathcal{N}(\mathbf{0}, \mathbf{I})$
2: $\boldsymbol{\tau}_T \leftarrow \tau_T(1 - m)$
3: $\mathbf{z}_{\boldsymbol{\tau}_T} = m\mathbf{z}_0^{\text{known}} + (1 - m)\boldsymbol{\epsilon}$
4: **for** $i = T, \ldots, 1$ **do**
5:                                            ▷ Partial DDIM sampling over unknown region
6:     $\hat{\boldsymbol{v}}_{\boldsymbol{\tau}_i} \leftarrow f_\theta(\mathbf{z}_{\boldsymbol{\tau}_i}, \boldsymbol{\tau}_i)$
7:     $\hat{\mathbf{z}}_0 \leftarrow \alpha_{\tau_i}\mathbf{z}_{\boldsymbol{\tau}_i} - \beta_{\tau_i}\hat{\boldsymbol{v}}_{\boldsymbol{\tau}_i}$
8:     $\hat{\boldsymbol{\epsilon}} \leftarrow \beta_{\tau_i}\mathbf{z}_{\boldsymbol{\tau}_i} + \alpha_{\tau_i}\hat{\boldsymbol{v}}_{\boldsymbol{\tau}_i}$
9:     $\hat{\mathbf{z}}_{\boldsymbol{\tau}_{i-1}}^{\text{unknown}} \leftarrow \alpha_{\tau_{i-1}}\hat{\mathbf{z}}_0 + \beta_{\tau_{i-1}}\hat{\boldsymbol{\epsilon}}$
10:     $\boldsymbol{\tau}_{i-1} \leftarrow \tau_{i-1}(1 - m)$
11:     $\mathbf{z}_{\boldsymbol{\tau}_{i-1}} \leftarrow m\mathbf{z}_0^{\text{known}} + (1 - m)\hat{\mathbf{z}}_{\boldsymbol{\tau}_{i-1}}^{\text{unknown}}$
12: **end for**
13: **return** $\mathbf{z}_{\boldsymbol{\tau}_0}$

---

as in Section 3.4. The model is trained using the velocity objective in Eq. (12), restricted to the unknown region. This allows the denoiser to explicitly learn the conditional score on $\mathbf{z}_\tau^\Gamma$, avoiding the need for stochastic marginalization and improving accuracy in conditional inpainting tasks. The full sampling procedure is detailed in Algorithm 2.

## C   Iterative Generation

In addition to the one-stage mixture generation described in Section 3.3, MGE-LDM also supports an iterative, stem-by-stem synthesis procedure. This approach constructs a full mixture by sequentially generating individual sources, leveraging the partial generation mechanism at each step.

Let $\mathcal{I} = \{c_i^{(s)}\}_i$ be an (ordered) set of CLAP embeddings corresponding to the desired instrument description. At the first iteration ($i = 1$), we generate an initial source latent $\hat{z}_1^{(s)}$ by sampling with the model conditioned only on the prompt $c_1^{(s)}$:

$$\tilde{z}^{(m)}, \tilde{z}^{(u)}, \hat{z}_1^{(s)} \sim p_\theta(z^{(m)}, z^{(u)}, z^{(s)} | \varnothing, \varnothing, c_1^{(s)}),$$

and set $z_1^{(u)} = \hat{z}_1^{(s)}$ as the initial submixture latent. For each subsequent iteration $i = 2, ..., |\mathcal{I}|$, we follow the iterative imputation strategy of partial generation, treating the current submixture as the accumulated sum of decoded sources from the previous steps.

Finally, the full mixture waveform is constructed by decoding and summing the generated source latents:

$$x^{(m)} = \sum_{i=1}^{|\mathcal{I}|} D(\hat{z}_i^{(s)}).$$

A preliminary evaluation of this iterative procedure is presented in Appendix E

## D   Experimental Details

This appendix provides a comprehensive description of datasets, baseline implementations, model architecture, and training settings used throughout our experiments.

### D.1   Datasets

We train and evaluate on three multi-track music datasets: Slakh2100 [37], MUSDB18 [38], and MoisesDB [39]. All datasets consist of mixture tracks paired with isolated stem recordings. A summary of stem abbreviations is provided in Table 4.

Table 4: **Abbreviations of instrument stems.** The table lists all abbreviations used throughout the paper along side their corresponding full instrument labels, grouped by dataset.

| Abbr. | Common | | | | | Slakh2100 | | | | | | | | MoisesDB | |
|---|---|---|---|---|---|---|---|---|---|---|---|---|---|---|---|
| | B | D | G | P | V | Brs. | C.P. | Org. | Pipe | Reed | Str. | S.Lead | S.Pad | B.str | Perc. |
| **Inst.** | bass | drums | guitar | piano | vocals | brass | chromatic percussion | organ | pipe | reed | strings | synth lead | synth pad | bowed strings | percussion |

**Slakh2100** is derived from the Lakh MIDI Dataset v0.1 [103] and contains synthesized tracks rendered with sample-based virtual instruments. It comprises 2100 songs divided into training (1500), validation (375), and test (225) splits, totaling approximately 145 hours of audio. It includes a wide variety of instrument classes (e.g., bass, drums, guitar, piano, strings, synth pad, etc.). We adopt the naming $\mathbf{S}_A$ to denote a subset containing only bass, drums, guitar, and piano, the four classes used by MSDM and MSG-LD, and $\mathbf{S}_B$ to denote the complementary subset of the remaining stems. We follow the official dataset splits provided by Slakh2100 for training, validation, and testing.

**MUSDB18** consists of 150 real-world music recordings with four stems: drums, bass, other, and vocals. We use all 100 tracks from the official training split for training and the 50-track test split for evaluation. The total dataset length is approximately 10 hours.

**MoisesDB** comprises 240 songs (14 hours total) contributed by 47 artists across 12 genres. Each stem in the song is annotated with a two-tier stem taxonomy. Each track is decomposed into its constituent sources and annotated using a two-level hierarchical taxonomy of stem classes. We aggregate all second-level tracks into their corresponding top-level classes. Among the 11 stem classes, we evaluate only the 7 unambiguous stems (e.g., bass, percussion, vocals, etc.). For evaluation, we randomly sample 24 tracks (10%) as the test set and use the remaining tracks for training.

**Data Construction.** We train our model using randomly constructed 3-track tuples (mix, sub, src). A source stem is randomly selected from the available stems, and the remaining stems are aggregated into a submixture. We select non-silent segments from the source track whenever possible, allowing up to 10 random resampling attempts per instance. The same temporal offset is applied across all stems to ensure alignment. For generation evaluation, we sample 300 random segments per test set. For source extraction, we sample between 150 and 700 non-silent segments per instrument class. All audio is downsampled to 16 kHz.

## D.2 Baseline Implementations

All baseline metrics are recomputed on our test splits for a fair comparison. The following models are used:

- MSDM [34]: We use the official implementation and pretrained checkpoint.[1] Since MSDM operates at 22 kHz, we upsample our 16 kHz test audio for inference and downsample the output back to 16 kHz.

- MSG-LD [36]: As no checkpoint is publicly released, we reproduce the model by retraining it from the official codebase.[2]

- HDemucs [20]: We train a 16 kHz version of Hybrid Demucs using the demucs_lightning implementation.[3]

- AudioSep [53]: We evaluate using the publicly available implementation and checkpoint provided by the authors.[4] Since AudioSep operates at a sampling rate of 32 kHz, we upsample all test audio from 16 kHz to 32 kHz before inference and subsequently downsample the separated outputs back to 16 kHz for evaluation consistency.

## D.3 Model Architecture and Training

**Autoencoder.** We adopt the VAE-based architecture from Stable Audio [72], with a downsampling ratio of 2048, yielding a 7.8125 Hz latent resolution and 64 latent channels. We train the autoencoder on all training subsets from Slakh2100, MUSDB18, and MoisesDB using 16 kHz mono audio for 600K steps with a batch size of 16.

**Diffusion Model.** In practice, the three latent representations are concatenated along the channel dimension, such that the input to the diffusion model becomes $Concat[z^{(m)}, z^{(u)}, z^{(s)}] \in \mathbb{R}^{3C \times L}$. We use a DiT backbone [86] with 24 transformer blocks with 48 heads, and a projected latent dimension of 1536 (3 tracks × 512

---

[1]https://github.com/gladia-research-group/multi-source-diffusion-models
[2]https://github.com/karchkha/MSG-LD
[3]https://github.com/KinWaiCheuk/demucs_lightning
[4]https://github.com/Audio-AGI/AudioSep

Table 5: **Source extraction performance of RePaint-based methods vs. MGE-LDM.** Metrics are reported as Log-Mel L1 distance $\downarrow$. $T$ indicates the number of reverse timesteps, and $U$ specifies the number of denoising operations per reverse step (i.e., $U-1$ intermediate resampling steps). The case $U = 1$ corresponds to the canonical single-sample conditional score approximation [11], as described in Appendix A.2. MGE-LDM uses adaptive timestep conditioning without resampling.

| Model | $T$ | $U$ | $\mathbf{S}_A$ | | | | | $\mathbf{S}_B$ | | | | | | | | |
|---|---|---|---|---|---|---|---|---|---|---|---|---|---|---|---|---|
| | | | B | D | G | P | Avg. | Brs. | C.P. | Org. | Pipe | Reed | Str. | S.Lead | S.Pad | Avg. |
| MGE (ours) | 250 | 1 | **1.68** | 2.71 | **2.69** | **2.16** | **2.31** | **3.43** | **2.16** | **1.84** | **2.33** | **3.07** | **2.44** | **2.31** | **1.93** | **2.48** |
| RePaint -based [89] | 250 | 1 | 2.00 | 3.20 | 3.15 | 2.83 | 2.79 | 4.75 | 2.47 | 2.79 | 4.27 | 4.65 | 3.06 | 3.73 | 2.80 | 3.56 |
| | 125 | 2 | 1.89 | 2.75 | 2.91 | 2.68 | 2.55 | 4.33 | 2.37 | 6.67 | 3.84 | 4.27 | 2.82 | 3.64 | 2.58 | 3.81 |
| | 50 | 5 | 1.80 | 2.43 | 2.83 | 2.55 | 2.40 | 3.89 | 2.32 | 2.57 | 3.35 | 3.81 | 2.64 | 3.56 | 2.42 | 3.07 |
| | 25 | 10 | 1.77 | **2.28** | 2.80 | 2.51 | 2.34 | 3.79 | 2.31 | 2.50 | 3.15 | 3.71 | 2.66 | 3.44 | 2.40 | 2.99 |
| | 250 | 2 | 1.90 | 2.65 | 2.94 | 2.65 | 2.53 | 4.23 | 2.41 | 2.71 | 3.82 | 4.27 | 2.84 | 3.66 | 2.59 | 3.31 |
| | 250 | 4 | 1.79 | 2.34 | 2.83 | 2.59 | 2.38 | 3.95 | 2.34 | 2.66 | 3.42 | 3.88 | 2.69 | 3.55 | 2.45 | 3.12 |

each). Timestep embeddings are prepended to the input vector of the transformer. CLAP embeddings for each track are processed by independent projection layers (without weight sharing) to produce scale and shift parameters for AdaIN-style conditioning [104]. These are applied group-wise via GroupNorm within each DiT layer to modulate the corresponding track-specific activations. Text embeddings are obtained from CLAP [56] using the checkpoint `music_audioset_epoch_15_esc_90.14.pt` via the `laion-clap` library.[5] Our implementation builds upon the official `stable-audio-tools` repository from Stability AI[6] and the training framework from `friendly-stable-audio-tools`.[7] All models were trained on a single NVIDIA RTX 6000 GPU (48 GB of memory).

# E  Ablation Study

This section presents ablation experiments designed to further analyze the key components of our framework. Unless otherwise specified, all models are trained on the combined $\mathbf{S}_A+\mathbf{S}_B$ dataset. Each of our models is evaluated with the same configuration as in Section 5, using $T = 250$ denoising steps during sampling.

## E.1  Comparison with Canonical Inpainting Methods

We assess the effectiveness of our adaptive timestep conditioning strategy by comparing it against two prior approaches: the canonical one-sample conditional score approximation (Appendix A.2) and the RePaint method [89] (Appendix A.3). Table 5 reports the results of the source extraction task.

In RePaint, $U$ denotes the number of denoising steps performed per reverse timestep: one denoising step followed by $U-1$ forward (resampling) steps. As a result, the total number of denoising steps becomes $T \times U$ during the full inpainting process. Note that setting $U = 1$ recovers the canonical single-sample estimator in Eq. (21).

We observe that, for a fixed number of denoising steps, using fewer timesteps $T$ with more resampling cycles $U$ generally improves performance, confirming observations in the original RePaint paper. We hypothesize that repeated resampling helps stabilize conditional generation by mitigating the noise mismatch between observed and unobserved regions, particularly at high noise levels, where observed latents contain little informative content and single-sample approximations of Eq. (21) become highly unreliable. While this approach does not yield a precise marginal score estimate, it heuristically improves inpainting quality through localized refinement.

Interestingly, we also observe that RePaint configurations with larger total denoising steps — such as $T = 250, U = 2$ and $T = 250, U = 4$ — consistently underperform compared to $T = 25, U = 10$, across all stems in both $\mathbf{S}_A$ and $\mathbf{S}_B$. This suggests that, for inpainting tasks, accurately modeling the conditional score at each timestep is more critical than simply increasing the number of reverse steps. As RePaint approximates the conditional score by marginalizing over perturbed conditions via resampling, performance benefits are observed primarily through increased resampling ($U$), not longer trajectories ($T$).

Nevertheless, our adaptive timestep model outperforms all RePaint variants across both datasets, with the sole exception of `drums` in $\mathbf{S}_A$. By directly learning track-specific conditional scores during training, our method eliminates the need for inference-time marginalization, resulting in lower variance and improved reconstruction quality.

---

[5] https://github.com/LAION-AI/CLAP
[6] https://github.com/Stability-AI/stable-audio-tools
[7] https://github.com/yukara-ikemiya/friendly-stable-audio-tools

Table 6: **Total generation performance across modeling variants.** Metrics are reported as FAD $\downarrow$. All models are trained on $\mathbf{S}_A + \mathbf{S}_B$. The baseline model uses uniform (non-adaptive) timesteps across all tracks. MGE variants apply adaptive timestep conditioning and test the impact of normalizations and CFG dropout rates. Values in parentheses indicate generation conditioned on the text prompt "*The sound of the bass, drums, guitar, and piano*".

| Model | Testset | | | |
|---|---|---|---|---|
| | $\mathbf{S}_A$ | $\mathbf{S}_{\text{Full}}$ | $\mathbf{M}_u$ | $\mathbf{M}_o$ |
| Non-adaptive | 3.26 (2.00) | 0.79 | **5.12** | **4.51** |
| MGE (adaptive) | 3.14 (2.24) | 0.63 | 5.46 | 4.73 |
|   - w/o GroupNorm | 3.48 (2.44) | 0.76 | 5.61 | 4.88 |
|   - CFG dropout $p=0.5$ | **3.12** (2.67) | **0.58** | 5.43 | 4.82 |

Table 7: **Source extraction performance under different architectural and CFG settings.** Metrics are reported as Log-Mel L1 distance $\downarrow$. All models are trained on $\mathbf{S}_A + \mathbf{S}_B$. GN and LN denote GroupNorm and LayerNorm, respectively. $p$ indicates the classifier-free guidance (CFG) dropout rate applied to each track's conditioning vector, and $s$ refers to the CFG guidance scale.

| Norm. | $p$ | $s$ | $\mathbf{S}_A$ | | | | | $\mathbf{S}_B$ | | | | | | | | |
|---|---|---|---|---|---|---|---|---|---|---|---|---|---|---|---|---|
| | | | B | D | G | P | Avg. | Brs. | C.P. | Org. | Pipe | Reed | Str. | S.Lead | S.Pad | Avg. |
| GN | 0.1 | 2.0 | 1.68 | 2.71 | 2.69 | 2.16 | 2.31 | 3.43 | **2.16** | **1.84** | 2.33 | 3.07 | 2.44 | 2.31 | 1.93 | 2.43 |
| LN | 0.1 | 2.0 | **1.67** | 4.22 | 2.65 | 2.15 | 2.67 | 3.42 | 2.35 | 1.97 | 2.40 | 3.45 | 2.51 | 2.32 | 2.03 | 2.55 |
| GN | 0.5 | 2.0 | 1.78 | **1.96** | 2.62 | 1.96 | 2.08 | 3.37 | 2.22 | 1.97 | 2.36 | 2.89 | 2.44 | **2.07** | 1.89 | 2.40 |
| GN | 0.1 | 1.0 | 1.67 | 2.79 | 2.70 | 2.05 | 2.30 | **3.24** | 2.23 | 1.85 | **2.25** | **2.88** | 2.28 | 2.27 | **1.87** | **2.35** |
| GN | 0.1 | 4.0 | 1.77 | 2.49 | 2.79 | 2.27 | 2.33 | 3.64 | 2.17 | 1.91 | 2.42 | 3.20 | 2.66 | 2.31 | 2.06 | 2.54 |
| GN | 0.1 | 8.0 | 1.93 | 2.49 | 2.93 | 2.41 | 2.44 | 4.35 | 2.28 | 2.01 | 2.57 | 3.47 | **3.27** | 2.42 | 2.30 | 2.83 |

A potential concern is whether optimizing for adaptive timestep-conditional inference might degrade generation quality when using uniform timestep schedules across tracks. To assess this, we evaluate our adaptive timestep model with a uniform timestep vector $\boldsymbol{\tau} = (\tau, \tau, \tau)$, which corresponds to the total generation task, and compare it to a baseline trained with non-adaptive, shared timesteps.

As shown in Table 6, comparing the non-adaptive uniform timestep baseline model and our model, both models achieve comparable FAD scores, indicating that timestep adaptation preserves generation performance under uniform scheduling while providing significant advantages for inpainting tasks.

## E.2 Additional Design Ablations

We additionally investigate the impact of various modeling and training choices, including normalization strategies and classifier-free guidance (CFG) dropout rates.

Table 6 includes results from models trained with LayerNorm instead of GroupNorm, following the original DiT architecture, as well as a variant using a higher CFG dropout rate of $p = 0.5$. We observe that GroupNorm slightly outperforms LayerNorm across all test sets, supporting the use of track-wise normalization in our multi-track setting. Regarding CFG dropout, increasing the dropout rate improves unconditional generation performance, particularly on $\mathbf{S}_A$ and $\mathbf{S}_B$. However, when conditioned on the text prompt (values in parentheses), the model trained with $p = 0.5$ performs worse, suggesting that overly aggressive dropout may impair semantic conditioning for total mixture generation.

We further examine how modeling and training design choices, such as normalization layers, classifier-free guidance (CFG) dropout probability, and CFG scale, affect extraction performance and report the results in Table 7. When comparing normalization strategies, GroupNorm consistently matches or outperforms LayerNorm across most stems, demonstrating the advantage of modeling track-wise statistics in our multi-track architecture. This observation aligns with the trends seen in mixture generation results. For CFG dropout, a higher dropout probability ($p = 0.5$) leads to improved performance compared to the default $p = 0.1$, suggesting that stronger stochastic conditioning is beneficial during source extraction. While this differs from the trend observed in mixture generation (Table 6), the discrepancy may be explained by the fact that, in extraction, non-target tracks are effectively treated as unconditioned. This makes overall performance more sensitive to the model's ability to generalize in the presence of dropout. We also evaluate various CFG scales (1, 2, 4, 8). A scale of 1 yields the best performance overall, although scales 2 and 4 remain competitive. Performance degrades at scale 8, indicating that overly strong guidance can impair extraction quality.

Table 8: **Preliminary results for iterative stem-wise generation.** Metrics are reported as FAD ↓. Evaluation is conducted using a model trained exclusively on $\mathbf{S}_A$. Each column header indicates the generation order of stems (e.g., BDGP denotes `bass→drums→guitar→piano`).

| | Total Gen. | BDGP | BDPG | DBGP | DBPG | DGBP | DPGB | GPBD | GPDB | PDGB | PGBD |
|---|---|---|---|---|---|---|---|---|---|---|---|
| | | | | | | $\mathbf{S}_A$ | | | | | |
| MGE | **0.47** | 0.60 | 0.66 | 0.78 | 0.75 | 0.70 | 0.77 | 0.61 | 0.66 | 0.63 | 0.60 |

Table 9: **Partial generation results evaluated with COCOLA [105] score ↑.** Higher values indicate stronger coherence between the given mixture and the generated stems, complementing *sub*-FAD by emphasizing musical consistency. "Ground truth" values correspond to the COCOLA scores computed between the given mixture and the ground-truth accompaniment from the dataset. Bold values indicate the best results in each column (excluding Ground truth), and underlined values denote the best results among models trained on the same $\mathbf{S}_A$ set.

| Model | | Train Set | | | | | | | | | | $\mathbf{S}_A$ | | | | | | | | | | $\mathbf{S}_B$ | | | | | |
|---|---|---|---|---|---|---|---|---|---|---|---|---|---|---|---|---|---|---|---|---|---|---|---|---|---|---|---|
| | | $\mathbf{S}_A$ | $\mathbf{S}_B$ | $\mathbf{M}_u$ | $\mathbf{M}_o$ | B | D | G | P | BD | BG | BP | DG | DP | GP | BDG | BDP | BGP | DGP | Brs. | C.P. | Org. | Pipe | Reed | Str. | S.Lead | S.Pad |
| MSDM | | ✓ | × | × | × | 57.74 | 51.87 | 57.76 | 56.10 | 45.96 | 57.64 | 54.48 | 50.67 | 49.40 | 55.75 | 43.25 | 40.46 | 52.82 | 47.09 | - | - | - | - | - | - | - | - |
| MSG-LD | | ✓ | × | × | × | 58.18 | 50.58 | 58.55 | 58.30 | 46.12 | 57.63 | 56.29 | 49.11 | 47.51 | 57.94 | 43.70 | 41.33 | 54.19 | 44.98 | - | - | - | - | - | - | - | - |
| MGE (ours) $\mathcal{T}_1$ | | ✓ | × | × | × | 60.93 | **53.71** | 58.09 | 59.78 | 49.77 | 60.43 | 59.16 | 52.95 | 51.90 | 60.03 | 46.28 | 43.46 | 56.77 | 49.73 | 63.20 | 56.44 | 59.52 | 60.59 | 62.35 | 61.11 | 63.74 | 58.66 |
| $\mathcal{T}_2$ | | ✓ | ✓ | × | × | **61.29** | 51.43 | **62.61** | **61.35** | 48.79 | **60.94** | 59.12 | 50.82 | 49.39 | **60.21** | 46.06 | 42.81 | 56.73 | 47.01 | 64.32 | **65.21** | **65.46** | 62.31 | **64.31** | 62.33 | 64.52 | **61.10** |
| $\mathcal{T}_3$ | | × | × | ✓ | ✓ | 55.40 | 44.76 | 57.29 | 59.34 | 43.90 | 55.18 | 56.09 | 45.68 | 45.42 | 55.91 | 44.10 | 43.35 | 51.44 | 43.53 | 60.23 | 54.74 | 56.70 | 60.18 | 56.50 | 60.66 | 62.03 | 61.00 |
| $\mathcal{T}_4$ | | ✓ | ✓ | ✓ | ✓ | 56.73 | 49.65 | 60.58 | 61.06 | 46.92 | 57.88 | 57.82 | 50.66 | 49.93 | 58.32 | **46.53** | **45.39** | 54.02 | 46.86 | **65.94** | 64.10 | 64.47 | **63.77** | 59.49 | **62.39** | 65.02 | 58.52 |
| Ground truth | | | | | | 60.11 | 53.55 | 56.33 | 57.16 | 50.42 | 59.74 | 58.59 | 53.11 | 52.29 | 57.80 | 47.06 | 44.32 | 56.07 | 50.18 | 64.58 | 64.34 | 64.85 | 63.38 | 64.75 | 63.20 | 65.29 | 62.16 |

## E.3 Iterative Generation Variants

Table 8 presents preliminary results for the iterative generation procedure described in Appendix C, applied to a model trained on $\mathbf{S}_A$. The task involves sequentially generating the four canonical stems (`bass`, `drums`, `guitar`, and `piano`) in various orders.

Across all tested permutations, iterative generation produced higher FAD scores compared to one-stage mixture generation, indicating a degradation in perceptual quality. Nonetheless, iterative generation may offer utility in settings that require fine-grained, source-specific control.

An interesting trend observed: generation sequences that began with `drums` consistently resulted in poorer performance relative to other orderings. This suggests that the model may be more effective at first establishing harmonic or melodic content before aligning rhythmic elements. While this observation is speculative, it highlights a potential inductive bias in the model that warrants further investigation, particularly in scenarios beyond the four-instrument configuration.

# F Extended Evaluation with Alternative Metrics

To complement the quantitative results presented in Section 5, we provide additional evaluations using alternative metrics that better capture musical coherence and perceptual quality. While *sub*-FAD is widely adopted in accompaniment generation research [34–36, 63, 76, 81], it is considered suboptimal for evaluating musical coherence, since it primarily measures global timbral similarity rather than the harmonic and rhythmic interplay between stems [105]. To address this limitation, Ciranni et al. [105] introduced the COCOLA metric, which quantifies harmonic and rhythmic coherence through contrastive learning of musical audio representations.

We report the COCOLA results in Table 9 to complement our main findings. The metric is computed between the given mixture and the generated stems, and we additionally report the "Ground truth" values computed between the same mixture and the corresponding reference accompaniment from the dataset. Among models trained on $\mathbf{S}_A$, our $\mathcal{T}_1$ variant outperforms the baselines on most instrument categories, except for guitar generation. Larger-scale models ($\mathcal{T}_2$–$\mathcal{T}_4$) show similarly strong or even superior coherence, in several cases approaching or surpassing ground-truth levels. Since COCOLA emphasizes musical coherence rather than timbral accuracy with respect to reference stems, it provides a meaningful complement to *sub*-FAD, offering a more balanced view of partial generation quality.

Because the VGGish-based FAD has been shown to correlate poorly with human perception in recent studies [106–108], we further report FAD$_{CLAP-MA}$ and *sub*-FAD$_{CLAP-MA}$, computed using CLAP embeddings from the `music_audioset_epoch_15_esc_90.14.pt` checkpoint, which achieves the highest correlation with subjective ratings according to Grötschla et al. [106]. Table 10 presents total generation results evaluated with FAD$_{CLAP-MA}$, where $\mathcal{T}_1$ achieves the best overall performance among models trained on $\mathbf{S}_A$. For the combined $\mathbf{S}_A+\mathbf{S}_B$ reference set, $\mathcal{T}_2$ yields the lowest FAD scores, while $\mathcal{T}_3$ performs best on $\mathbf{M}_u$ and $\mathbf{M}_o$, aligning with

Table 10: **Total generation results evaluated with FAD$_{CLAP\text{-}MA}$ ↓.**

| Model | | Train Set | | | | Test Set | | |
|---|---|---|---|---|---|---|---|---|
| | | $\mathbf{S}_A$ | $\mathbf{S}_B$ | $\mathbf{M}_u$ | $\mathbf{M}_o$ | $\mathbf{S}_A$ | $\mathbf{S}_{\text{Full}}$ | $\mathbf{M}_u$ | $\mathbf{M}_o$ |
| MSDM | | ✓ | × | × | × | .329 | .290 | .767 | .744 |
| MSG-LD | | ✓ | × | × | × | .191 | .312 | .658 | .697 |
| MGE (ours) | $\mathcal{T}_1$ | ✓ | × | × | × | **.110** (.450) | .285 | .637 | .627 |
| | $\mathcal{T}_2$ | ✓ | ✓ | × | × | .368 (.359) | **.099** | .578 | .621 |
| | $\mathcal{T}_3$ | × | × | ✓ | ✓ | .731 (.669) | .632 | **.222** | **.165** |
| | $\mathcal{T}_4$ | ✓ | ✓ | ✓ | ✓ | .563 (.712) | .426 | .235 | .186 |

Table 11: **Partial generation results evaluated with *sub*-FAD$_{CLAP\text{-}MA}$ ↓.**

| Model | | Train Set | | | | $\mathbf{S}_A$ | | | | | | | | | | | | | | $\mathbf{S}_B$ | | | | | | | |
|---|---|---|---|---|---|---|---|---|---|---|---|---|---|---|---|---|---|---|---|---|---|---|---|---|---|---|---|
| | | $S_A$ | $S_B$ | $M_u$ | $M_o$ | B | D | G | P | BD | BG | BP | DG | DP | GP | BDG | BDP | BGP | DGP | Brs. | C.P. | Org. | Pipe | Reed | Str. | S.Lead | S.Pad |
| MSDM | | ✓ | × | × | × | .103 | .100 | **.066** | .096 | .227 | .189 | .252 | .135 | .191 | .302 | .275 | .357 | .555 | .403 | - | - | - | - | - | - | - | - |
| MSG-LD | | ✓ | × | × | × | **.071** | **.070** | .076 | **.077** | **.130** | **.135** | **.143** | **.117** | **.121** | **.157** | **.185** | **.188** | .241 | **.190** | - | - | - | - | - | - | - | - |
| MGE (ours) | $\mathcal{T}_1$ | ✓ | × | × | × | .140 | .170 | .153 | .154 | .184 | .175 | .181 | .200 | .189 | .184 | .221 | .218 | **.190** | .214 | **.278** | .086 | .069 | .327 | .462 | .201 | .335 | .118 |
| | $\mathcal{T}_2$ | ✓ | ✓ | × | × | .259 | .329 | .251 | .264 | .420 | .302 | .362 | .380 | .361 | .337 | .500 | .517 | .422 | .460 | .425 | **.071** | .129 | .327 | .330 | .296 | .123 | .313 |
| | $\mathcal{T}_3$ | × | × | ✓ | ✓ | .178 | .202 | .356 | .253 | .307 | .402 | .340 | .368 | .320 | .477 | .445 | .469 | .632 | .484 | .361 | .087 | **.062** | **.045** | **.090** | **.087** | **.079** | **.060** |
| | $\mathcal{T}_4$ | ✓ | ✓ | ✓ | ✓ | .170 | .187 | .367 | .251 | .253 | .405 | .308 | .350 | .289 | .455 | .450 | .436 | .571 | .486 | .312 | .072 | .107 | .057 | .340 | .208 | .085 | .063 |

their respective training domains. The *sub*-FAD$_{CLAP\text{-}MA}$ results for partial-generation, presented in Table 11, exhibit trends consistent with Table 2, with MSG-LD achieving the best scores on most stem combinations. Together, these additional metrics provide a more comprehensive and reliable assessment of both perceptual quality and cross-domain generalization in partial and total music generation.

# G  Future Work

Future extensions of MGE-LDM include scaling to higher-resolution formats such as 44.1 kHz stereo audio, enabling richer timbral detail and spatial fidelity. In particular, this can be achieved by leveraging high-quality latent representations recently developed for the music domain [109, 110]. To reduce the modality gap in text-conditioned extraction, fine-tuning on curated audio–text datasets like MusicCaps [7] is a promising direction. Given its minimal reliance on precise stem boundaries, MGE-LDM is naturally suited for incorporating weakly or noisily labeled multi-track data [39, 111], which may expand training diversity.

Another promising avenue is to pre-train MGE-LDM on large-scale mixture-only corpora such as MTG-Jamendo [112] or the Free Music Archive [113] to learn general audio priors for mixture tracks, followed by fine-tuning on multi-track datasets for source-aware generation. This two-stage training strategy is expected to enhance generative quality and improve generalization.

We also plan to extend MGE-LDM to text-based music editing tasks, drawing inspiration from recent instruction-guided frameworks such as AUDIT [73], InstructME [74], and Instruct-MusicGen [64]. Leveraging MGE-LDM's latent inpainting capabilities and language-conditioned generation, this extension could enable user-directed operations such as instrument replacement and style transformation via natural language prompts, building upon the model's unified training scheme and class-agnostic design.

# H  Spectrogram Examples of Generated Samples

We present Mel-spectrogram visualizations of generated audio samples across the three primary tasks: total generation, partial generation (imputation), and source extraction. All examples are produced by MGE-LDM trained on the combined Slakh2100 ($\mathbf{S}_A$+$\mathbf{S}_B$), MUSDB18 ($\mathbf{M}_u$), and MoisesDB ($\mathbf{M}_o$) datasets.

We note that the model is capable of generating `vocals` in the unconditional setting, as `vocals` stems are present in the training data. Although MGE-LDM does not currently support fine-grained control over vocal generation, this points to a promising direction for future work, such as incorporating explicit vocal prompts or segment-level control for more expressive and structured multi-track modeling.

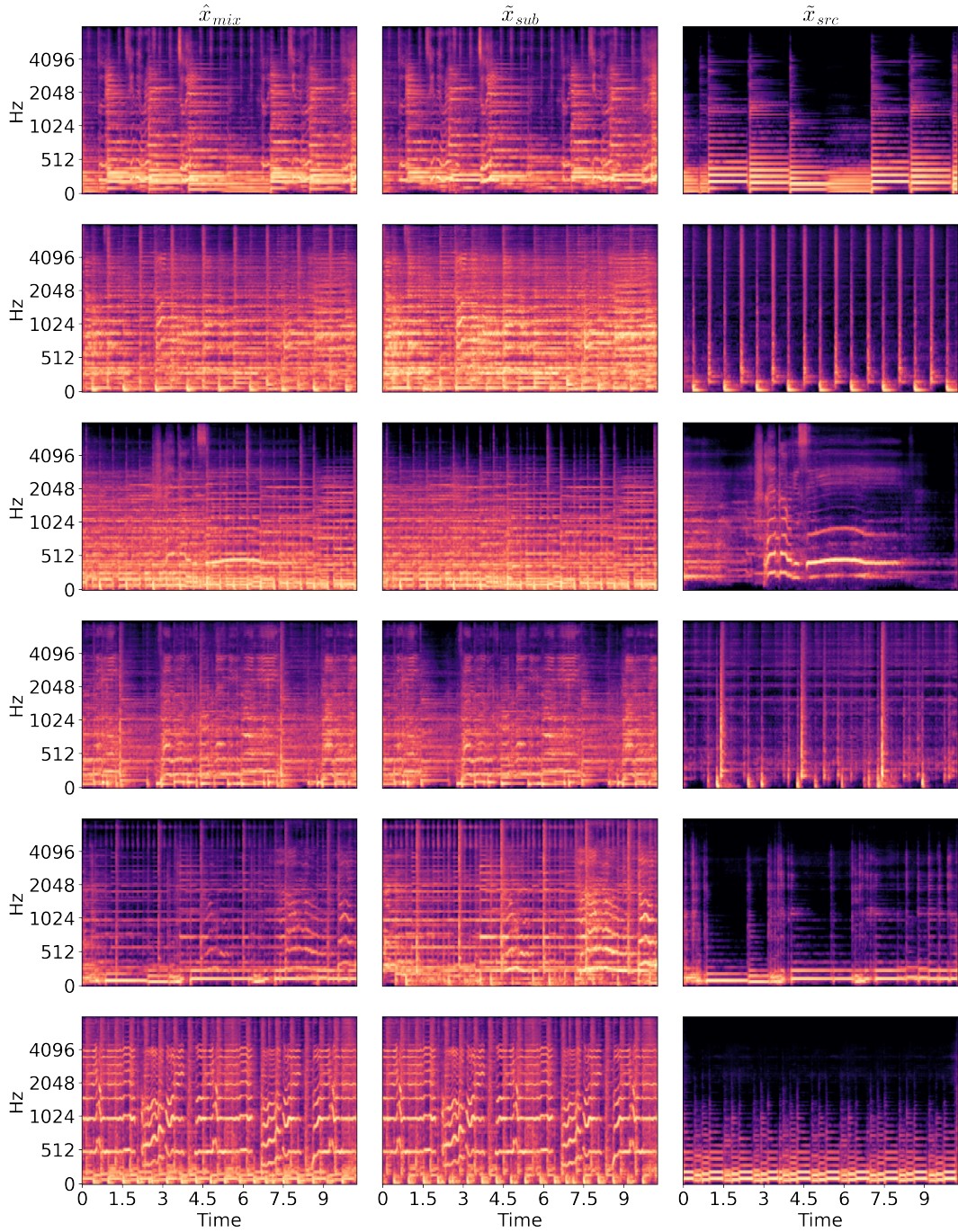

Figure 2: **Total generation examples.** Each sample displays Mel-spectrograms of the mixture, submixture, and source tracks, all generated simultaneously by MGE-LDM. The mixture track is used to evaluate the total generation output.

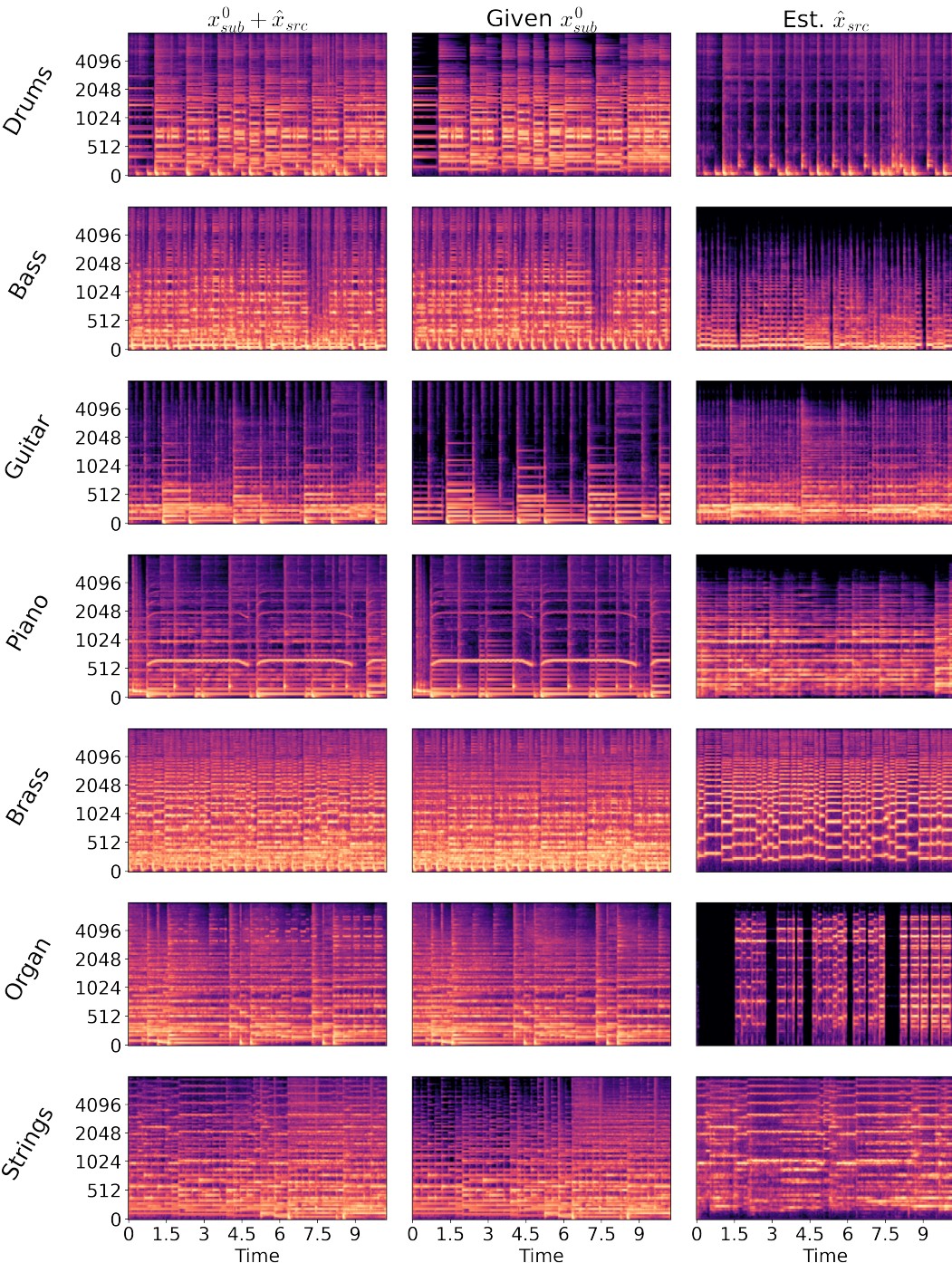

Figure 3: **Source imputation examples.** Each row illustrates source inpainting results by MGE-LDM, conditioned on the text prompt "*The sound of the* {label}". The middle column shows the provided context mixture (submix), the rightmost column is the generated source, and the leftmost column is the recombined mixture of the submix and generated source. While some stems are imputed accurately, others fail due to data imbalance during training.

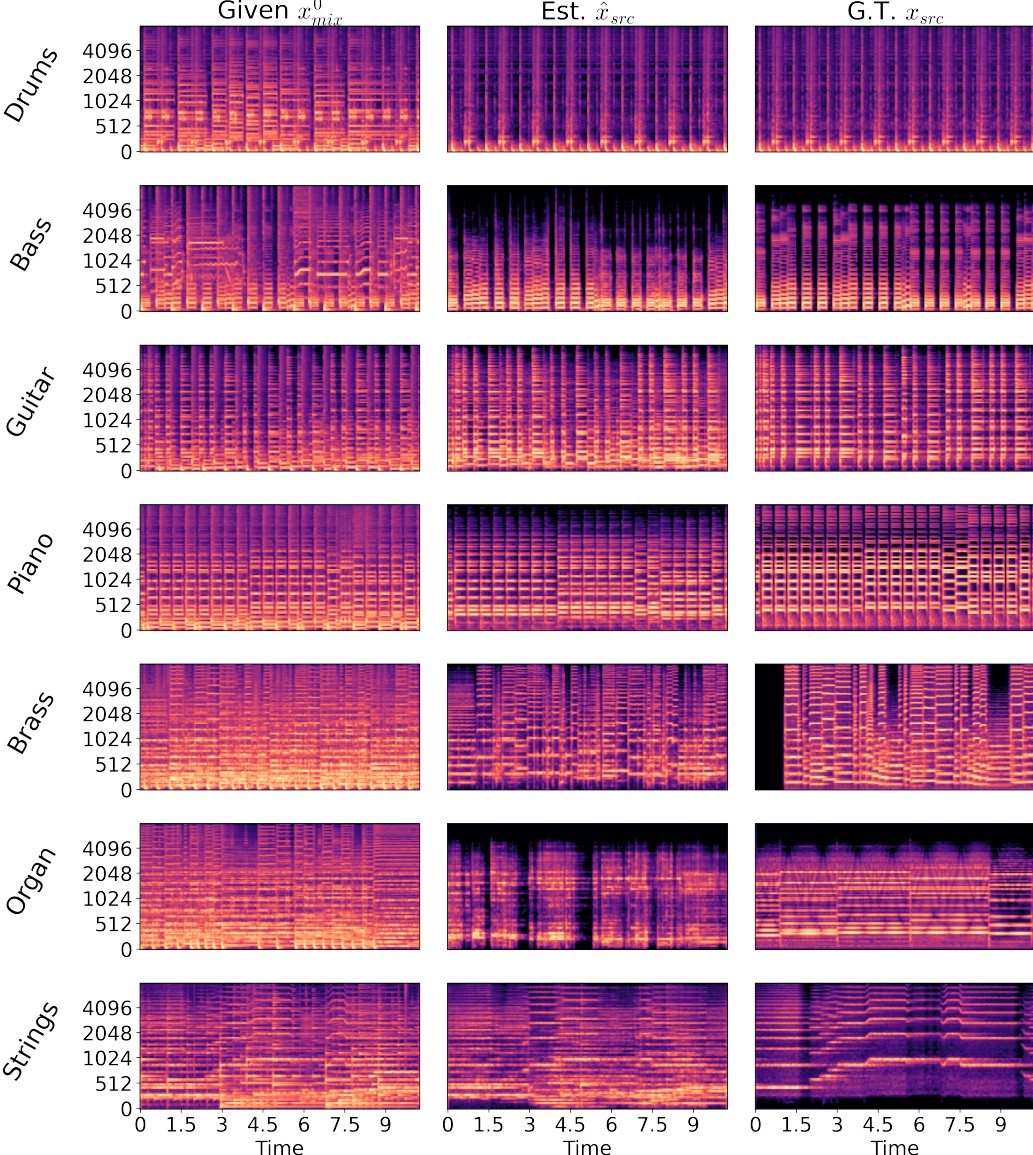

Figure 4: **Source extraction examples.** Source extraction results produced by MGE-LDM, conditioned on the text query "*The sound of the* {label}". The leftmost column shows the input mixture, the middle column is the extracted source predicted by the model, and the rightmost column is the ground-truth source. We observe that extraction quality may degrade for underrepresented classes such as strings, and in some cases, the model hallucinates unrelated instruments or incorrect timbres.

# I   Ethics Statement

This work introduces a class-agnostic generative framework for multi-track music modeling, trained exclusively on publicly available datasets (Slakh2100, MUSDB18, and MoisesDB). While the model enables flexible music generation, source imputation, and source extraction, it also carries potential risks, such as unauthorized manipulation, misuse in derivative content, or generation of audio resembling copyrighted material. To mitigate these concerns, we commit to releasing the model and code under a license with clear usage guidelines, emphasizing responsible research and ethical creative applications.

