# OpenReview forum: "MGE-LDM: Joint Latent Diffusion for Simultaneous Music Generation and Source Extraction"
_NeurIPS.cc/2025/Conference — NeurIPS 2025 poster_

### Official Review · Reviewer_cGKq · 2025-06-20

**Clarity:** 2
**Significance:** 2
**Originality:** 2
**Rating:** 3
**Confidence:** 4

**Summary:**

The authors introduce MGE-LDM, a unified latent diffusion framework for simultaneous music generation, source imputation, and text-driven source separation. Unlike prior methods that depend on fixed instrument classes or assume additive mixing in the waveform domain, MGE-LDM operates in a shared latent space and models the joint distribution over mixtures, submixtures, and individual sources. By formulating both stem completion and source extraction as conditional inpainting tasks and leveraging CLAP embeddings for text conditioning, the model enables class-agnostic manipulation of arbitrary instruments.

**Questions:**

- Why are the authors only training on 16 kHz? As far as I know the Stable Audio pipeline supports up to 44.1 kHz stereo.
- Why are the authors using an outdated version of Demucs? The Hybrid Demucs is significantly better compared to the original.
- Why does the demo page only have comparisons in the Slakh section but not for Moises/MusDB? Especially for the source extraction, which can be easily compared to existing methods.

**Ethical Concerns:**

["NO or VERY MINOR ethics concerns only"]

**Final Justification:**

Even though paper does not show substantial improvements over prior work, the approach contains novelty and should be interesting to the audience of NeurIPS.

**Limitations:**

The authors have a section on limitations in the appendix. I suggest they move this to the main text.

**Quality:**

2

**Strengths And Weaknesses:**

While none of the individual components of the proposed architecture are novel, the combination of diffusing stems in the latent space based on clap conditioning signals (and therefore not requiring explicit stem labels) seems novel.


Unfortunately, I’m not convinced by the results. The empirical study using VGGish FAD is poor, as multiple previous works have shown that VGGish is a suboptimal embedding model for music [1-3]. Additionally, while I appreciate the authors having trained various models (T1-4), I don’t see how it makes sense to compare these against a baseline only trained on a single subset of the data. While the results on synthetic data (slakh) seem decent, the model underperforms when listening to real music (moises, musdb). The biggest issue I have is the missing human evaluation, as the FAD numbers don’t show the full picture.


[1] Gui, Azalea, et al. "Adapting frechet audio distance for generative music evaluation." ICASSP 2024-2024 IEEE International Conference on Acoustics, Speech and Signal Processing (ICASSP). IEEE, 2024.


[2] Grötschla, Florian, et al. "Benchmarking Music Generation Models and Metrics via Human Preference Studies." ICASSP 2025-2025 IEEE International Conference on Acoustics, Speech and Signal Processing (ICASSP). IEEE, 2025.

[3] Huang, Yichen, et al. "Aligning Text-to-Music Evaluation with Human Preferences." arXiv preprint arXiv:2503.16669 (2025).

---

> ### Author Rebuttal · Authors · 2025-07-31
>
> We sincerely thank Reviewer cGKq for your careful reading and constructive comments. Below, we address your main concerns:
>
> > *"Unfortunately, I’m not convinced by the results. The empirical study using VGGish FAD is poor, as multiple previous works have shown that VGGish is a suboptimal embedding model for music [1-3]."*
>
> **Authors' Response**:
> We appreciate your feedback and agree that VGGish-based FAD is not the ideal metric for music generation.
> We initially adopted it for its widespread use as in SingSong [4] and MSDM [5], but we acknowledge its limitations.
> We additionally calculated the FAD using CLAP-MA embeddings (LAION-CLAP checkpoint `music_audioset_epoch_15_esc_90.14`), which Grötschla et al. [2] have shown best correlation with human perception.
>
> Below, Table A reports the total-generation results using FAD-CLAP-MA.
>
> ---
>
> *Table A: Total generation result with FAD-CLAP-MA $\downarrow$*
> |Model | Trainset | $S_A$ | $S_{full}$ | $M_u$ | $M_o$ |
> | ---------- | -----| ------- | -------- | -------- | -------- |
> MSDM | $S_A$ | 0.329 | 0.290 | 0.767 | 0.744 |
> MSG-LD | $S_A$ | 0.191 | 0.312 | 0.658 | 0.697 |
> $T_1$ | $S_A$ | **0.110** (0.450) | 0.285 | 0.637 | 0.627 |
> $T_2$ | $S_A$+$S_B$ | 0.368 (0.359) | **0.099** | 0.578 | 0.621 |
> $T_3$ | $M_u$+$M_o$ | 0.731 (0.669) | 0.632 | **0.222** | **0.165** |
> $T_4$ | $S_A$+$S_B$+$M_u$+$M_o$ | 0.563 (0.712) | 0.426 | 0.235 | 0.186 |
>
> ---
>
> It shows a similar relative ranking to Table 1 of the main paper, with lower absolute values.
> One notable difference is that our $T_1$ model achieves the lowest FAD-CLAP-MA score compared to the baselines, which is a clear contrast to the VGGish-based results in Table 1 of the main paper.
>
> We also computed the FAD-CLAP-MA for partial generation task, where the results are shown below in Table B.
>
> ---
>
> *Table B: Partial generation result with sub-FAD-CLAP-MA $\downarrow$*
>
> |Model | Trainset | B | D | G | P | BD | BG | BP | DG | DP | GP | BDG | BDP | BGP | DGP |
> | ------- | -----| ------- | -------- | -------- | -------- | -------- | -------- | -------- | -------- | -------- | -------- | -------- | -------- | -------- | -------- |
> | MSDM | $S_A$ | 0.103 | 0.100 | **0.066** | 0.096 | 0.227 | 0.189 | 0.252 | 0.135 | 0.191 | 0.302 | 0.275 | 0.357 | 0.555 | 0.403 |
> | MSG-LD | $S_A$ | **0.071** | **0.070** | 0.076 | **0.077** | **0.130** | **0.135** | **0.143** | **0.117** | **0.121** | **0.157** | **0.185** | **0.188** | 0.241 | 0.190 |
> | $T_1$ | $S_A$ | 0.140 | 0.170 | 0.153 | 0.154 | 0.184 | 0.175 | 0.181 | 0.200 | 0.189 | 0.184 | 0.221 | 0.218 | **0.190** | 0.214 |
> | $T_2$ | $S_A$+$S_B$ | 0.259 | 0.329 | 0.251 | 0.264 | 0.420 | 0.302 | 0.362 | 0.380 | 0.361 | 0.337 | 0.500 | 0.517 | 0.422 | 0.460 |
> | $T_3$ | $M_u$+$M_o$ | 0.178 | 0.202 | 0.356 | 0.253 | 0.307 | 0.402 | 0.340 | 0.368 | 0.320 | 0.477 | 0.445 | 0.469 | 0.632 | 0.484 |
> | $T_4$ | $S_A$+$S_B$+$M_u$+$M_o$ | 0.170 | 0.187 | 0.367 | 0.251 | 0.253 | 0.405 | 0.308 | 0.350 | 0.289 | 0.455 | 0.450 | 0.436 | 0.571 | 0.486 |
> ---
> ---
> ---
>
> |Model| Trainset | Brass | C.P. | Org. | Pipe | Reed | Str. | S.Lead | S.Pad |
> | :---------- | :--------- | --------: | ----------: | --------: | --------: | --------: | -------: | ---------: | --------: |
> | $T_1$ | $S_A$  | **0.278** |  0.086 | 0.069 | 0.327 | 0.462 | 0.201 |   0.335 | 0.118 |
> | $T_2$ | $S_A$+$S_B$ | 0.425 |   **0.071** | 0.129 | 0.327 | 0.330 | 0.296 |   0.123 |     0.313 |
> | $T_3$ | $M_u$+$M_o$ | 0.361 |   0.087 | **0.062** | **0.045** | **0.090** | **0.087** |   **0.079** |  **0.060** |
> | $T_4$ | $S_A$+$S_B$+$M_u$+$M_o$ | 0.312 |   0.072 | 0.107 | 0.057 | 0.340 | 0.208 |   0.085 |     0.063 |
> ---
>
>
> The overall relative pattern of the Table B closely matches Table 2 of the main text, with MSG-LD achieving the best scores on most stem combinations.
> We believe these additional metrics offer a more reliable assessment of partial generation performance.
>
> We also agree that FAD alone cannot capture musical coherence, so we also evaluated the COCOLA score [6] for partial generation ***(please refer to the table and discussion in our response to Reviewer W8on)***.
>
>
> > *"Additionally, while I appreciate the authors having trained various models (T1-4), I don’t see how it makes sense to compare these against a baseline only trained on a single subset of the data."*
>
> **Authors' Response**:
> You are correct that only the baselines (MSDM and MSG-LD) and our $T_1$, each trained on the same $S_A$ set, are directly comparable.
> We included $T_2$~$T_4$ to illustrate how training on larger and more diverse datasets affects performance, since dataset-agnostic scaling is rarely evaluated in prior work and no existing baselines cover these settings.
> We hope these results will serve as useful reference points for future research.
>
> For additional comparability, we also trained Hybrid Demucs on the $S_A$ set. The corresponding results are shown in Table C below to strenghten the baseline comparison.
>
> ---
>
> *Table C: Source extraction results with Hybrid Demucs.
> Column headers indicate the target stem and its corresponding set.*
>
> |Model| Trainset |B ($S_A$) | D ($S_A$) | G ($S_A$) | P ($S_A$) | V ($M_u$) | B ($M_u$) | D ($M_u$) | V ($M_o$) | B ($M_o$) | D ($M_o$) | G ($M_o$) | P ($M_o$) |
> | ------- | -----| ------- | -------- | -------- | -------- | -------- | -------- | -------- | -------- | -------- | -------- | -------- | -------- |
> |Demucs | $M_u$ | 1.29 | 0.92 | - | - | 1.59 | **1.67** | 1.29 | 0.89 | **1.48** | 1.08 | - | - |
> |HDemucs | $M_u$ | 1.49 | 0.90 | - | - | **1.50** | 1.99 | 1.53 | **0.83** | 1.71 | 1.1 | - | - |
> |HDemucs | $S_A$ | **0.83** | **0.57** | **0.78** | **0.79** | - | 1.87 | **1.22** | - | 1.61 | **1.01** | **1.45** | **0.99** |
> |$T_1$ 	| $S_A$ | 1.28 | 0.66 | 1.27 | 1.07 | 3.80 | 1.91 | 1.33 | 5.15 | 1.61 | 1.10 | 2.86 | 2.68 |
>
> ---
>
> > *"The biggest issue I have is the missing human evaluation, as the FAD numbers don’t show the full picture."*
>
> **Authors' Response**:
> We agree that human evaluation is crucial for assessing generated music quality.
> Accordingly, we conducted a listening study on a total generation task, and the results are shown in Table D below.
>
> ---
>
> *Table D: Mean Opinion Score (MOS) for the total‐generation task, rated from 1 (bad) to 5 (excellent). Two aspects were evaluated: Quality (how much the snippet sounds like noise versus a real song) and Coherence (consistency across generated stems). A total of 24 subjects participated, each listening to 8 segments per model. Scores are reported with 95% confidence intervals.*
>
> |Model | Ground truth | MSDM | MSG-LD | $T_1$ | $T_2$ | $T_3$ | $T_4$ |
> | ------ | ----- | ----- | ------ | ------- | ------- | ----- | ------- |
> |Quality| 3.79±0.18 | 2.05±0.16 | 3.80±0.17 | 3.63±0.17 | 3.92±0.16 | 3.81±0.16 | 3.92±0.16 |
> |Coherence| 3.75±0.19 | 2.06±0.18 | 3.70±0.17 | 3.54±0.18 | 3.99±0.15 | 3.75±0.17 | 3.79±0.15 |
> ---
>
>
> The results indicate that $T_1$ is slightly less preferred than MSG-LD, whereas $T_2$~$T_4$ receive higher preference ratings, demonstrating that expanding the training dataset can benefit the perceptual quality of generated music.
>
>
> > Q:*"Why are the authors only training on 16 kHz? As far as I know the Stable Audio pipeline supports up to 44.1 kHz stereo."*
>
> **Authors’ Response:**
> We initially chose 16 kHz mono to match the baseline models and ensure a fair comparison. We agree that 16 kHz mono is not sufficient for high-fidelity music generation (as noted in our Limitations section). Having demonstrated the validity of our framework at 16 kHz, we plan to extend MGE-LDM to 44.1 kHz stereo in future work.
>
> > Q:*"Why are the authors using an outdated version of Demucs? The Hybrid Demucs is significantly better compared to the original."*
>
> **Authors’ Response:**
> Thank you for pointing this out.
> We have attached results for Hybrid Demucs (HDemucs) in *Table C* above.
> All models were trained to convergence.
> In our experiments, HDemucs results have not shown significant improvements over standard Demucs, so we initially reported the original Demucs comparison in the main paper, consistent with the MSDM paper. However, we agree that including HDemucs in the main tables is more appropriate and will update the manuscript accordingly.
>
>
> > Q:*"Why does the demo page only have comparisons in the Slakh section but not for Moises/MusDB? Especially for the source extraction, which can be easily compared to existing methods."*
>
> **Authors’ Response:**
> Thank you for your feedback regarding the demo page.
> On the “Slakh Full” demo page, we initially showcased only $T_4$ results, assuming that adding other references would not be informative given the diversity of instruments.
> However, we agree that this limits comparability.
> Accordingly, we have now added Demucs inference examples alongside our model under “Source Extraction (Full) / MUSDB18 & Moises".
> We hope this makes the demo more informative and directly comparable.
>
> > *"The authors have a section on limitations in the appendix. I suggest they move this to the main text."*
> Thank you for the suggestion. We will move the Limitations section into the main text to ensure it is clearly visible to readers.
>
> We appreciate your valuable feedback and will incorporate these changes in the final manuscript to enhance its clarity and rigor.
>
>
> **References**
>
> [1] Gui et al. "Adapting frechet audio distance for generative music evaluation." ICASSP 2024
>
> [2] Grötschla et al. "Benchmarking Music Generation Models and Metrics via Human Preference Studies." ICASSP 2025
>
> [3] Huang et al. "Aligning Text-to-Music Evaluation with Human Preferences." arXiv:2503.16669, 2025.
>
> [4] Donahue et al. "Singsong: Generating musical
> accompaniments from singing." arXiv:2301.12662, 2023.
>
> [5] Mariani et al. "Multi-source diffusion models for simultaneous music generation and separation." ICLR 2024
>
> [6] Ciranni et al. "Cocola: Coherence-oriented contrastive learning of musical audio representations." ICASSP 2025.

---

> > ### Comment · Reviewer_cGKq · 2025-08-01
> >
> > I thank the authors for their effort made in responding to my concerns.
> >
> > Reading the response I would like to highlight two aspects:
> > 1. The authors agree, that the only fair comparison between the proposed method and previous work/baselines is $T_1$. Therefore, these tables should be split as $T_2$ through $T_4$ should be treated more as an ablation on how the performance increases when more & varied data is added. For a truly fair comparison $T_2$ through $T_4$ would have to be compared with previous methods trained on the same data.
> > 2. Looking at the comparison of $T_1$, it outperforms previous approaches for the subset-generation (bass, drums, guitar, and piano) in FAD, it underperforms for the full generation in FAD, it slightly underperforms in the human evaluation of the total generation, it underperforms according to the partial generation results, and it underperforms against HDemucs baseline on separation results. Is my understanding in this regard correct?

---

> > > ### Author Response · Authors · 2025-08-02
> > > **Response to Reviewer cGKq**
> > >
> > > Thank you for your comments and your careful review of our rebuttal.
> > >
> > > 1. As you suggested, we will present $T_1$ results alone alongside the baselines for a fair comparison, and report $T_2$-$T_4$ in a separate table. We will explicitly note in the manuscript and table captions that the $T_2$-$T_4$ results serve purely to evaluate the impact of dataset scaling.
> > >
> > > 2. You are correct in your summary. However, if "full generation" refers to specifically to the $FAD_{CLAP}$ result for $S_{full}$ in the total-generation task, $T_1$ actually outperforms the baseline. We will restate the exact performance breakdown as you summarized in the manuscript to make these trade-offs clear.
> > >
> > > We appreciate your guidance and will incorporate these revisions into the final draft.

---

> > > > ### Comment · Reviewer_cGKq · 2025-08-02
> > > >
> > > > I thank the authors for their clarifying statements. I have no further questions at this time.

---

> > > > > ### Author Response · Authors · 2025-08-05
> > > > >
> > > > > Thank you for your time and careful consideration. We appreciate your feedback, and please don’t hesitate to reach out if you have any further questions.

---

> ### Comment · Reviewer_W8on · 2025-08-06
>
> The reviewer should re-consider their evaluation, especially after additional quantitative evaluation on COCOLA score, which complements the sub-FAD evaluation (and offers a more important metric for capturing coherence on the partial generation task). Also, using VGGish embeddings for sub-FAD is not really poor. **Practically all the field of accompaniment generation (SingSong [1], MSDM [2], GMSDI [3], COCOLA [4], MSG-LD [5], JEN-1 Composer [6], STAGE [7]) uses VGGish embeddings for sub-FAD!**. Also with the authors providing metrics with CLAP embeddings and giving a more detailed picture, the current experimental results are more than sufficient.
>
> References
> - [1] Donahue, Chris, et al. "Singsong: Generating musical accompaniments from singing." arXiv preprint arXiv:2301.12662 (2023).
> - [2] Mariani, Giorgio, et al. "Multi-Source Diffusion Models for Simultaneous Music Generation and Separation." The Twelfth International Conference on Learning Representations.
> - [3] Postolache, Emilian, et al. "Generalized multi-source inference for text conditioned music diffusion models." ICASSP 2024-2024 IEEE International Conference on Acoustics, Speech and Signal Processing (ICASSP). IEEE, 2024.
> - [4] Ciranni, Ruben, et al. "Cocola: Coherence-oriented contrastive learning of musical audio representations." ICASSP 2025.
> - [5] Karchkhadze, Tornike, Mohammad Rasool Izadi, and Shlomo Dubnov. "Simultaneous music separation and generation using multi-track latent diffusion models." ICASSP 2025-2025 IEEE International Conference on Acoustics, Speech and Signal Processing (ICASSP). IEEE, 2025.
> - [6] Yao, Yao, et al. "Jen-1 composer: A unified framework for high-fidelity multi-track music generation." Proceedings of the AAAI Conference on Artificial Intelligence. Vol. 39. No. 13. 2025.
> - [7] Strano, Giorgio, et al. "STAGE: Stemmed Accompaniment Generation through Prefix-Based Conditioning." arXiv preprint arXiv:2504.05690 (2025).

---

> ### Comment · Reviewer_cGKq · 2025-08-08
>
> Dear reviewer W8on,
>
> The provided experimental results are indeed more than sufficient, I have not said anything to the contrary. However, from how I see it, the results don't show outperformance of prior work to warrant acceptance. The results from COCOLA look promising, but the rebuttal is not a place to meticulously inspect and re-evaluate new results. For example. how can we be sure there was no inadvertent information leakage, as COCOLA trained on many of the same datasets that were also used in this work. All the provided results together give an impression of not significantly better than previous work, especially the human evaluation which is the gold-standard metric. Therefore, I politely disagree with your assessment and maintain my original score.
>
> On the matter of VGGish. SingSong started with VGGish at a time when this drawback was not known yet, and then every other paper followed by reusing it. Does this mean it's a good embedding model? I would argue not. Especially since there is now ample evidence to support that VGGish is a poor choice for music models.
>
> We can gladly continue this discussion during the AC-reviewer phase.

---

### Official Review · Reviewer_W8on · 2025-06-21

**Clarity:** 3
**Significance:** 3
**Originality:** 3
**Rating:** 5
**Confidence:** 5

**Summary:**

This paper follows the line of research set up by MSDM (Mariani et al. 2024), where a diffusion model is trained on stem-separated musical data (of fixed instrument classes) and is able to perform simultaneously full track generation, accompaniment generation given conditioning tracks and source separation. MSDM was generalised in GMSDI (Postolache et al. 2024), where text conditioning is introduced and by MSG-LD (Karchkhadze et al. 2025), which sets the diffusion model in a latent space, improving quality. MSG-LD though keeps the number of instrument classes fixed like MSDM and cannot generate arbitrary classes via text conditioning. Doing all tasks in the latent domain is difficult especially because of source separation, which requires linearity in the likelihood function (that's why GMSDI works in the waveform domain only). In this paper, the authors propose MGE-LDM, a method that aims to improve on both limitations, defining a latent diffusion model (via Stable Audio Open) that can be text conditioned and can solve all three tasks simultaneously. They train over triples composed by $(z^{(m)}, z^{(u)}, z^{(s)})$, where $z^{(m)}$ is a latent mixture, $z^{(u)}$ is a latent sub-mixture (the mixture minus a source) and $z^{(s)}$ is a source. At the same time they condition on CLAP embeddings of the form $(c^{(m)}, c^{(u)}, c^{(s)})$. The three tasks at inference time:
- Total generation: sample the triple and decode $z^{(m)}$.
- Partial generation: inpaint $(z^{(m)}), z^{(s)})$ given $z^{(u)}$ and decode $z^{(s)}$.
- Source separation: inpaint $(z^{(u)}), z^{(s)})$ given $z^{(m)}$ and decode $z^{(s)}$.

The authors also introduce track-individual noising in order to improve the inpainting mechanism. The authors benchmark their model against MSDM and MSG-LD reaching good results (especially on source separation), with improved generalisability of the model.

**References**
- Mariani, Giorgio, et al. "Multi-Source Diffusion Models for Simultaneous Music Generation and Separation." ICLR 2024.
- Postolache, Emilian, et al. "Generalized multi-source inference for text conditioned music diffusion models." ICASSP 2024.
- Karchkhadze, Tornike, Mohammad Rasool Izadi, and Shlomo Dubnov. "Simultaneous music separation and generation using multi-track latent diffusion models." ICASSP 2025.

**Questions:**

- Line 162: Do we have a different $p$ on each $c^{(k)}$?
- Line 176: I would say: "whose score we approximate with $f_\theta$".
- Line 270: "Results" instead of "Result"
- Line 293: This makes sense since FAD cannot capture the distribution shift.
- Line 297: I think the authors could improve on these results with negative conditioning of unwanted tracks (see GMSDI, Fig. 2).
- Line 630: Before giving the expression of  the velocity, the expression of $z_t$ in function of $\epsilon$ and $z_0$ should be given.
- Line 631: The $0$ subscript should be not bold.

**Ethical Concerns:**

["NO or VERY MINOR ethics concerns only"]

**Final Justification:**

I keep my full accept on this paper! This is a great improvement in the area of compositional audio generation, finally having a well-defined stem-level generative model in a latent domain. I am glad the authors introduced evaluations on COCOLA score, showcasing improved stem coherence with respect to previous baselines.

**Limitations:**

yes

**Quality:**

3

**Strengths And Weaknesses:**

**Strengths**

- It is interesting to see the multi-track music generation research direction being improved, especially given the presented model it is general enough to let one train over multiple stem-separated datasets at the same time. This is a great departure from models with fixed classes (MSDM, MSG-LD).

- Quantiative evaluation is thorough, comparing to MSDM and MSG-LD in a plethora of tasks. Would have been cool to compare to GMSDI given the commonality of text conditioning, but such paper did not release code / checkpoints so I agree it is a thorough task. (still one key metric is missing for partial generation, see Weaknesses #1)

- The track-aware inpainting method is very interesting. Also seeing the comparison with RePaint in Table 5 of Appendix D and seeing the presented inpainting method as the best performing, gives a strong indicative that this should be the default mechanism for multi-track generation (at least in the absence of a constraining latent likelihood, see Weaknesses #2).

- Cool to see a discussion about iterative generation in Appendix C. Having the individual separated tracks of a mixed track was one of the rationales of  MSDM so it is interesting to see one is not losing that feature (it is actually generalised).

- Overall the paper is great to read and a good contribution in the (controllable) music generation domain.

**Weaknesses**


- As the author acknowledge in the limitations section, we still depend on labels in a loose way when training.

- A limitation of the paper is that partial generation metrics are computed on sub-FAD, a metric which, while having served well as an initial foray in evaluating accompaniments models (e..g, in SingSong, in MSDM, etc.), there is empirical evidence (Ciranni et al. 2025) that it is not optimal for such a task, favouring randomly mixed stems over natural combinations. The authors should also compute metrics such as COCOLA score (Ciranni et al. 2025), which are better suited for the task. Also in that paper one finds a benchmark on a conditional model based on Stable Audio Open, released publicly, and trained on MusDB and MoisesDB, which shares the diffusion architecture of the presented paper: it would be interesting to see how the presented joint model compares to the conditional one.

- While the authors acknowledge that the nonlinearity of the latent space makes it difficult to constrain the sources in a mixture and they opt out for their approach, I believe that the three entries of a triple are not necessarily constrained to sum well in the data (waveform) domain. This could have a negative impact in the performance of the model. Thus, finding the best way to solve compositional music generation in the latent domain it still remains an open problem, and maybe a more constrained approach would be given, for example, by learning latent-domain likelihood functions like in (Postolache et al. 2023). The authors could discuss this in the concluding remarks of the paper or in limitations.

**References**

- Ciranni, Ruben, et al. "Cocola: Coherence-oriented contrastive learning of musical audio representations." ICASSP 2025.
- Postolache, Emilian, et al. "Latent autoregressive source separation." Proceedings of the AAAI Conference on Artificial Intelligence. Vol. 37. No. 8. 2023.

---

> ### Author Rebuttal · Authors · 2025-07-30
>
> We sincerely thank Reviewer W8on for the through and insightful feedback.
> We appreciate your recognition of our contritbutions to multi-track music generation, and we address your comments below.
>
>
> > *"A limitation of the paper is that partial generation metrics are computed on sub-FAD, a metric which, while having served well as an initial foray in evaluating accompaniments models (e..g, in SingSong, in MSDM, etc.), there is empirical evidence (Ciranni et al. 2025) that it is not optimal for such a task, favouring randomly mixed stems over natural combinations. The authors should also compute metrics such as COCOLA score (Ciranni et al. 2025), which are better suited for the task. Also in that paper one finds a benchmark on a conditional model based on Stable Audio Open, released publicly, and trained on MusDB and MoisesDB, which shares the diffusion architecture of the presented paper: it would be interesting to see how the presented joint model compares to the conditional one."*
>
> **Authors' Response**:
> You're correct that sub-FAD alone can bias toward random mixtures rather than musical coherence.
> In response to your suggestion, we computed the COCOLA score [1] on exactly the same partial-generation outputs reported in Table2 of the main paper.
> The results are shown below, with the highest COCOLA score per column bolded, and among models trained on the $\mathbf{S}_A$ set (MSDM, MSG-LD, or our T1), the best result underlined.
>
> | Model | Trainset        |    **B** |  **D** |     **G** |     **P** |    **BD** |    **BG** | **BP** |    **DG** |    **DP** |    **GP** |   **BDG** |   **BDP** |   **BGP** |   **DGP** |
> | ---------------- | -------- | -----------: | --------: | --------: | --------: | --------: | --------: | --------: | --------: | --------: | --------: | --------: | --------: | --------: | --------: |
> | MSDM             | $\textbf{S}_A$         | $57.74$ |     $51.87$ |     $57.06$ |     $56.10$ |     $45.96$ |     $57.64$ |     $54.48$ |     $50.67$ |     $49.40$ |     $55.75$ |     $43.25$ |     $40.46$ |     $52.82$ |     $47.09$ |
> | MSG‑LD           | $\textbf{S}_A$         |        $58.18$ |     $50.58$ |     $\underline{58.55}$ |     $58.30$ |     $46.12$ |     $57.63$ |     $56.29$ |     $49.11$ |     $47.51$ |     $57.94$ |     $43.70$ |     $41.33$ |     $54.19$ |     $44.98$ |
> | $\mathcal{T}_1$   | $\textbf{S}_A$         |        $\underline{60.93}$ | $\underline{\mathbf{53.71}}$ |     $58.09$ |     $\underline{59.78}$ |     $\underline{\mathbf{49.77}}$ |     $\underline{60.43}$ | $\underline{\mathbf{59.16}}$ | $\underline{\mathbf{52.95}}$ |     $\underline{\mathbf{51.90}}$ |     $\underline{60.03}$ |     $\underline{46.28}$ |     $\underline{43.46}$ | $\underline{\mathbf{56.77}}$ |     $\underline{\mathbf{49.73}}$ |
> |
> | $\mathcal{T}_2$   | $\textbf{S}_A$+$\textbf{S}_B$      |    $\mathbf{61.29}$ |     $51.43$ | $\mathbf{62.61}$ | $\mathbf{61.35}$ |     $48.79$ | $\mathbf{60.94}$ |     $59.12$ |     $50.82$ |     $49.39$ | $\mathbf{60.21}$ |     $46.06$ |     $42.81$ |     $56.73$ |     $47.01$ |
> | $\mathcal{T}_3$        | $\textbf{M}_u$+$\textbf{M}_o$    |        $55.40$ |     $44.76$ |     $57.29$ |     $59.34$ |     $43.90$ |     $55.18$ |     $56.09$ |     $45.68$ |     $45.42$ |     $55.91$ |     $44.10$ |     $43.35$ |     $51.44$ |     $43.53$ |
> | $\mathcal{T}_4$       | $\textbf{S}_A$+$\textbf{S}_B$ +$\textbf{M}_u$+$\textbf{M}_o$|        $56.73$ |     $49.65$ |     $60.58$ |     $61.06$ |     $46.92$ |     $57.88$ |     $57.82$ |     $50.66$ |     $49.93$ |     $58.32$ |     $\mathbf{46.53}$ |     $\mathbf{45.39}$ |     54.02 |     46.86 |
> |
> | **Ground Truth** | –          |        $60.11$ |     $53.55$ |     $56.33$ |     $57.16$ | $50.42$ |     $59.74$ |     $58.59$ | $53.11$ | $52.29$ |     57.80 | $47.06$ | $44.32$ |     $56.07$ | $50.18$ |
>
> ---
> ---
> ---
> ---
>
> | Model            | Trainset                                           | **Brs.** | **C.P.** | **Org.** |  **Pipe** |  **Reed** | **Str.** | **S.Lead** | **S.Pad** |
> | :--------------- | :------------------------------------------------- | --------: | ----------: | --------: | --------: | --------: | ----------: | ---------------: | -------------: |
> | T1    | $\mathbf{S}_A$  | $63.20$ |   $56.44$ | $59.52$ | $60.59$ | $62.35$ | $61.11$ |   $63.74$ | $58.66$ |
> | T2  |$\textbf{S}_A$+$\textbf{S}_B$ | $64.32$ |   $\textbf{65.21}$ | $\textbf{65.46}$ | $62.31$ | $\textbf{64.31}$ | $62.33$ |   $64.52$ |     $\textbf{61.10}$ |
> | T3 | $\mathbf{M}_u$+$\mathbf{M}_o$ | $60.23$ |   $54.74$ | $56.70$ | $60.18$ | $56.50$ | $60.66$ |   $62.03$ |  $61.00$ |
> | T4  | $\textbf{S}_A$+$\textbf{S}_B$+$\mathbf{M}_u$+$\mathbf{M}_o$ | $\textbf{65.94}$ |   $64.10$ | $64.47$ | $\textbf{63.77}$ | $59.49$ |  $\textbf{62.39}$ |   $\textbf{65.02}$ |     $58.52$ |
> |
> | **Ground Truth** | –  | $64.58$ |   $64.34$ | $64.85$ | $63.38$ | $64.75$ |  $63.20$ |   $65.29$ |     $62.16$ |
>
> ---
>
> The results show that our MGE-LDM variants achieve the highest coherence across nearly all stem combination compared to the baselines (MSDM and MSG-LD), often matching or even approaching the ground-truth levels.
>
> Because COCOLA emphasizes musical coherence rather than timbral accuracy to reference stems, we believe it complements sub-FAD and provides a more balanced view of partial generation quality.
> We will integrate these COCOLA results either alongside sub-FAD in the main manuscript or in the Appendix, as you suggested.
>
> (For reference, the COCOLA score (Harmonic) of stable audio + control net reported by the COCOLA paper [1] is 57.34, though it is not directly comparable to our results since it uses a different evaluation set.)
>
> Once again, thank you for this valuable suggestion.
>
> > *"While the authors acknowledge that the nonlinearity of the latent space makes it difficult to constrain the sources in a mixture and they opt out for their approach, I believe that the three entries of a triple are not necessarily constrained to sum well in the data (waveform) domain.
> This could have a negative impact in the performance of the model.
> Thus, finding the best way to solve compositional music generation in the latent domain it still remains an open problem, and maybe a more constrained approach would be given, for example, by learning latent-domain likelihood functions like in (Postolache et al. 2023).
> The authors could discuss this in the concluding remarks of the paper or in limitations."*
>
> **Authors' Response**:
> You rightly point out that, although we train on triplets satisfying $mix = submix + source$ in waveform space, our latent-diffusion model does not enforce an explicit additivity constraint for generated triplets.
> This issue is critical, as we believe it directly related to the “hallucination” phenomenon, in which the model extracts sources that aren’t actually present in the mixture.
> Postolache et al. [2] offer an elegant solution by enforcing additivity in a discrete VQ-VAE latent space:
> they estimate the joint likelihood of two sources by counting exact codebook occurences (bin-counting), effectively modeling $p(z_{mix}|z_{src_1}, z_{src_2})$, where $z$ are quantized latent codes.
> This bin-counting approach serves as a powerful regularizer to enforce a summation constraint in the latent domain.
> Unfortunately, our current pipeline relies on a continuous latent space, which precludes direct application of discrete bin counts.
>
> Nonetheless, we believe adapting this latent-domain likelihood approach to our model would be highly promising.
> We could either extend it to a continuous latent space by designing appropriate regularizers, or convert our encoder to a discrete representation, such as using VQ-VAE based encoder and discrete diffusion (or MaskGIT) for generation to enable bin counting in the latent space.
>
> We thank the reviewer for this valuable suggestion, and we will discuss these directions in the Limitations and Future Work sections as promising approaches to reduce hallucinations and improve compositional consistency.
>
> We also appreciate your helpful line-by-line suggestions and will update the manuscript accordingly.
> In particular, your idea of negative conditioning to supress unwanted tracks is very valuable and will be explored in our future work.
>
> **References**:
>
> [1] Ciranni, Ruben, et al. "Cocola: Coherence-oriented contrastive learning of musical audio representations." ICASSP 2025.
>
> [2] Postolache, Emilian, et al. "Latent autoregressive source separation." AAAI 2023.

---

> > ### Author Response · Authors · 2025-08-05
> >
> > Dear Reviewer W8on,
> >
> > I hope you are doing well. With the author-reviewer discussion period extended until August 8, I wanted to gently follow up on our rebuttal, in which we addressed your insightful questions. We would greatly appreciate any further feedback or clarifications you might have at your convenience. Thank you again for your time and thoughtful review.

---

> > ### Comment · Reviewer_W8on · 2025-08-06
> >
> > I am sorry for my late answer! I deeply appreciate the authors have found time to integrate the COCOLA score in their manuscript. It definitely improves the quality of their submission, achieving such strong results on the metric. Also discussing latent consistency in the manuscript is a great addition. I keep my full accept score!

---

> > > ### Author Response · Authors · 2025-08-07
> > >
> > > We are sincerely grateful for the time you devoted to reviewing our rebuttal, for your constructive and insightful feedback, and for your full-accept recommendation. We will incorporate your suggestions into the revised manuscript.

---

### Official Review · Reviewer_ShCW · 2025-06-24

**Clarity:** 3
**Significance:** 3
**Originality:** 3
**Rating:** 5
**Confidence:** 4

**Summary:**

The paper introduces MGE-LDM, a latent diffusion model that simultaneously addresses music generation, source imputation, and text-driven source separation without relying on predefined instrument classes. Unlike prior models constrained to fixed stem definitions, MGE-LDM learns a joint distribution over mixtures, submixtures, and individual sources in a compressed latent space. By framing both source extraction and stem completion as conditional inpainting tasks, the model can handle arbitrary instruments and enables language-guided manipulation using CLAP embeddings. It further introduces track-wise adaptive timestep conditioning during training to enhance inpainting fidelity, allowing the model to distinguish between observed and missing components. Empirical evaluation demonstrates competitive performance across synthetic and real-world benchmarks in total generation, partial generation, and source extraction; an accompanying demo website confirms the competitiveness of the approach.

**Questions:**

- Can you quantify or mitigate the limitations of CLAP-based conditioning? CLAP is conceptually appealing for language-guided extraction, but its granularity and semantic specificity are limited. Do you observe consistent issues with ambiguous or overlapping prompts (e.g., "strings" vs. "synth pad")? Have you explored alternative or complementary embeddings, or prompt disambiguation strategies?

- What are the primary failure modes, and how often do they occur?

**Ethical Concerns:**

["NO or VERY MINOR ethics concerns only"]

**Final Justification:**

The authors clarified my concerns and included additional results that I explicitly asked for. Having also gone through the other reviews + rebuttal, I can see that several other experiments address all the other major concerns raised by the other reviewers. I think the authors did an excellent job for this rebuttal. Therefore, I confirm my recommendation to accept the paper for presentation at NeurIPS.

**Limitations:**

Yes.

**Paper Formatting Concerns:**

None.

**Quality:**

3

**Strengths And Weaknesses:**

This paper presents a timely contribution to the field of music generation and source separation by proposing a latent diffusion model that, similarly to recent methods, unifies multiple tasks in a coherent framework. The technical foundation is solid, and empirically the model is clearly capable of high-fidelity generation and source manipulation, with performance that is competitive with or better than strong baselines. One of the key strengths lies in its use of submixtures as an intermediate representation, which elegantly works around the need to know the exact number or type of stems present in a mixture. This design choice improves scalability and robustness, especially when working with heterogeneous real-world datasets where source labels are often noisy or aggregated. The use of CLAP embeddings for conditioning is another good idea. While CLAP is not flawless, its integration here is conceptually compelling. Finally, I appreciated the adaptation of TD-Paint's per-region timestep conditioning to the audio domain, which is both technically interesting and practically useful in improving latent inpainting fidelity.

An aspect I find worthy of praise is the possibility to move beyond the common and somewhat restrictive four-stem modeling (bass, drums, guitar, piano/other) seen in many prior works. By demonstrating the model's ability to handle a wider range of instruments including synths, brass, pipe, and others, it takes an important step toward more realistic and musically rich applications, even if the results for these less common stems are not yet exceptional. This direction is significant because it reflects the diversity of real-world music and shows a commitment to broader applicability.

The paper is generally well-organized, though some sections (e.g. the method) can be dense and may benefit from additional high-level intuition or summarization. Nonetheless, the figures and detailed mathematical exposition help make the approach reproducible. The joint modeling of mixture, submixture, and source in the latent space, combined with conditional inpainting, represents a meaningful departure from previous work that tends to rely on fixed-class assumptions or waveform-domain additivity.

Key weaknesses include limitations in generalization and fidelity for non-canonical instruments, but these are also clearly outlined by the authors. This suggests the model struggles with finer details or less frequent classes, and points to room for improvement in generalization to rare or underrepresented sources. But I don't see this as a critical drawback. Further, as already mentioned, while the method is mathematically rigorous, its clarity suffers from heavy notation and long explanations without always grounding the reader in intuitive takeaways. The presentation could be tightened to emphasize conceptual contributions before diving into formalism.

Overall, I believe the paper meets the bar for NeurIPS, given its improvements upon prior related work published in such venues.

---

> ### Author Rebuttal · Authors · 2025-07-29
>
> We thank Reviewer ShCW for the thoughtful and constructive feedback and are delighted that you recognize our work as a timely contribution to the field. Below, we address each of your points:
>
> > *"The paper is generally well-organized, though some sections (e.g. the method) can be dense and may benefit from additional high-level intuition or summarization. /  while the method is mathematically rigorous, its clarity suffers from heavy notation and long explanations without always grounding the reader in intuitive takeaways. The presentation could be tightened to emphasize conceptual contributions before diving into formalism."*
>
> **Authors' response:**
> We agree that the Method section can feel dense. To make it more accessible, we will revise the method section to include a conscise, high-level overview outlining the key ideas in plain language before describing mathematical details.
>
> > *"Q: Can you quantify or mitigate the limitations of CLAP-based conditioning? CLAP is conceptually appealing for language-guided extraction, but its granularity and semantic specificity are limited. Do you observe consistent issues with ambiguous or overlapping prompts (e.g., "strings" vs. "synth pad")? Have you explored alternative or complementary embeddings, or prompt disambiguation strategies?"*
>
> **Authors' response:**
> You’re correct that CLAP’s semantic granularity can lead to ambiguity.
> To illustrate, we ran a small experiment on mixtures containing strings or synth‑pad and measured Log‑Mel L1 distances between the true isolated stems and the model’s output under different text prompts:
>
> | Cond. inst. \ Ref inst. | strings | synth_pad |
> |:------------------------|--------:|----------:|
> | strings                 |    2.57 |      2.12 |
> | synth_pad               |    2.25 |      2.06 |
>
> In an ideal setting, the lowest errors would lie on the diagonal, but here we see that asking for “synth pad” on a mixture with strings yields a lower L1 distance (2.25) than asking for “strings” (2.57).
> Listening confirms that the model often reproduces the correct pitch contour but with a synth‑like timbre, which is a hallucinatory behavior we have noted in our Limitations section. This highlights an important direction for future work in improving prompt‑based source separation (which we also discuss with the reviewer W8on).
>
> We also intend to explore more recent embedding models such as M2D‑CLAP [1] to enhance semantic precision, investigate negative prompting strategies (e.g. “strings, not synth pad”) to explicitly suppress unwanted classes.
> We also can try leverage CLAP’s audio‑branch embeddings from short reference clips to anchor the model and reduce ambiguity. Although we have not yet quantified these ideas, we believe they could meaningfully improve disambiguation.
>
>
> > *"Q: What are the primary failure modes, and how often do they occur?"*
>
> **Authors' response:**
> Empirically, we observe three primary failure modes.
> First, the model sometimes fails to extract a source and outputs near‑silence instead of the intended stem.
> Second, with ambiguous prompts the model often hallucinates sounds that aren’t present in the mixture, as discussed above. This issue seldom arises for clear prompts like “the sound of vocals,” where the model correctly outputs silence when vocals are absent.
> Third, the model struggles to capture the fine-grained detail of isolated single notes on piano and guitar: while chords and percussive strokes are generally well reproduced, individual notes often vanish or become blurred.
> We believe these issues stem from a relative scarcity of fine-detail, single-source examples in our training data.
> To address them, we plan to augment our dataset with additional isolated‑note samples and experiment with increased loss weighting on fine‑detail reconstruction.
>
>
> **Reference**
>
> [1] Niizumi, D., et al. “M2D‑CLAP: Masked modeling duo meets CLAP for learning general‑purpose audio‑language representations.” Interspeech 2024.

---

> > ### Comment · Reviewer_ShCW · 2025-08-01
> >
> > I thank the authors for the additional clarifications and experiments. I still do recommend acceptance of this paper.

---

> > > ### Author Response · Authors · 2025-08-02
> > > **Response to Reviewer ShCW**
> > >
> > > We would like to express our sincere gratitude again for your constructive and insightful feedback. We will incorporate your suggestions into the revised version.

---

### Note · Authors · 2025-08-11

We sincerely thank all reviewers for their careful reading and constructive feedback, and the AC for facilitating the discussion and shepherding the process.

We acknowledge that, in fair same-dataset comparisons, our results are competitive rather than uniformly superior to baselines. However, we believe that the core contribution of this work lies in a **methodological advance in generality and scalability**, which broadens what is possible in multi-track music generation/separation research.

Our framework removes fixed-stem assumptions via a joint latent modeling strategy that casts separation and imputation as conditional diffusion inpainting, enabling class-agnostic scaling to heterogeneous multi-track datasets without predefined instrument categories; we further improve inpainting with adaptive timestep conditioning.

We believe these properties open a path beyond previous fixed-class, dataset-specific pipelines and expand what the community can build in multi-track music generation and separation.

We respectfully ask the AC and reviewers to weigh this generality and scalability, and practical flexibility alongside the reported metrics. Thank you again for the thoughtful consideration.

---

### Decision · Program_Chairs · 2025-09-17

**Decision:**

Accept (poster)

**Comment:**

The paper proposes MGE-LDM, a latent diffusion model for multi-track music generation, source imputation, and text-conditioned source separation without relying on fixed stem categories. The framework models mixtures, sub-mixtures, and sources jointly in a latent space and frames both extraction and completion as conditional inpainting, with adaptive timestep conditioning to improve fidelity.

All reviewers agree on the significance of the methodological contribution, which allows handling mixtures beyond the standard four-stem pipelines and offers a scalable, stem-agnostic approach. The ability to exploit heterogeneous datasets during training is also an important contribution of this work.

One reviewer remains unconvinced by the experimental evaluation, noting that the proposed method does not always outperform competitors, particularly in human evaluation. This concern was partially addressed by the introduction of the COCOLA metric.

Even if the performance is not always superior, the clear methodological advance, the ability to unify multiple tasks in a scalable latent framework, and the new experiments presented during the rebuttal make this work a clear step forward in this line of research.